# Inverse design of structural colours in polymeric films with crystallization-induced reversible thermochromism

Dong Yang[1], Heyi Liang[2] ✉, Chengjie Zhang[1], Peipei Shao[3], Qin Li[1], Yun Huang[1], Yi Dan[1], Cheng Zeng [4], Rui-Tao Wen [3], Long Jiang [1] ✉ & Ming Xiao [1] ✉

Precisely controlling structural colours in polymeric materials remains a major challenge, with current approaches often relying on trial-and-error synthesis. Here, we develop a colour design model, enabling inverse design of structural colours in bottlebrush block copolymers (BBCPs). The model can quantitatively link BBCP molecular structures to macroscopic colours through the integration of a strong segregation self-consistent field theory model with a multilayer optical framework. We first validate its predictive capability by synthesising and assembling BBCPs with varied chain architectures to produce a full colour spectrum, and then demonstrate its generalisability to other BBCP chemistries. In addition, we observe reversible, nonlinear thermochromism in systems combining a crystalisable block with a soft, low–glass transition temperature segment, while similar BBCPs lacking this pairing show no such response. Our work establishes a predictive platform for designing structurally coloured, thermoresponsive polymeric materials and advances the rational engineering of photonic soft matter.

Inverse design is revolutionising material discovery by shifting trial-and-error synthesis to a predictive framework that identifies optimal material structures based on target properties[1]. A quantitative structure-property relationship makes inverse design more efficient and predictable. In polymeric materials, significant progress has been made in correlating molecular structures with bulk mechanical properties, such as the influence of polymer topology on elasticity[2,3]. However, predicting bulk optical properties, such as colours, remains challenging due to the complex interplay between chromophores[4,5], polymeric chain interactions[6,7], and nanoscale structures[8,9]. Establishing robust structure-property relationships for optical properties is crucial for advancing inverse design in polymeric systems, enabling precise control for applications in display, photonic coating, anti-counterfeiting, and camouflage.

Structural colours, known for their vibrancy, durability, and environmental friendliness, have attracted considerable attention[10,11]. Bottlebrush block copolymers (BBCPs), which are macromolecules with densely grafted polymeric side chains[12,13], offer a promising platform for generating such colours. These polymers can self-assemble into well-ordered nanostructures with domain sizes of hundreds of nanometres, resulting in structural colours[14–16]. Achieving such large domain sizes is kinetically challenging in ultrahigh linear block copolymers due to severe entanglements[17]. Since the first report of blue structural colour in self-assembled polystyrene-*block*-polylactide BBCPs[18], a broad spectrum of colours has been achieved by precisely controlling narrow-dispersity, high molecular weight, and chemistry of BBCPs through advanced synthesis methods[19,20]. Beyond film-based applications, BBCPs have also been used as photonic resins or inks in extrusion[21] and injection-based 3D colour printing[22,23]. Additionally, the

[1]College of Polymer Science and Engineering, Polymer Research Institute, State Key Laboratory of Advanced Polymer Materials, Sichuan University, Chengdu, China. [2]Pritzker School of Molecular Engineering, University of Chicago, Chicago, Illinois, USA. [3]Department of Materials Science and Engineering, Southern University of Science and Technology, Shenzhen, China. [4]Key Laboratory of Multifunctional Nanomaterials and Smart Systems, Suzhou Institute of Nano-Tech and Nano-Bionics, Chinese Academy of Sciences, Suzhou, China. ✉e-mail: heyi@uchicago.edu; jianglong@scu.edu.cn; mingxiao@scu.edu.cn

self-assembly of amphiphilic BBCPs in emulsion droplets has enabled the fabrication of 3D porous photonic balls with tunable structural colours[24–26].

Despite advances in BBCP-based structural colours, no predictive model has yet quantitatively linked BBCP molecular structures to their macroscopic colours. Particle-based simulations, such as atomistic and coarse-grained simulations[27,28], are computationally expensive for BBCPs with domain sizes exceeding 100 nm due to high molecular weight. Field-based simulations, including self-consistent field theory (SCFT)[29,30] and field-theoretic simulations[31], enable larger-scale modelling by representing polymers as density fields rather than particles. However, incorporating chemically specific molecular details requires complex particle-to-field transformations[32]. Additionally, most models neglect the effects of crystallisation, which can impart new functionalities into structurally coloured materials[33]. More importantly, existing polymer physics models can at most predict nanostructures from molecular architectures but offer no link to colours, limiting their applicability to inverse design.

To address these challenges, we here generalise an analytic model based on strong segregation self-consistent field (SS-SCF) theory[33,34] to predict the domain spacing of nanostructures in self-assembled BBCPs with chemically distinct backbones and side chains[35]. By integrating this polymer physics model with a multilayer optical framework, we develop a colour design model that directly links chain architectures to macroscopic colours. To validate this model, we synthesise poly-dimethylsiloxane-*block*-poly (ethylene glycol) (PDMS-*b*-PEO) with different chain architectures and assemble them to produce a full colour spectrum. The model prediction not only closely aligns with experimental results in the melt state, but also accurately predicts the crystallised state by incorporating a chain stiffening parameter to account for PEO block crystallisation. Extending this framework to other monomer chemistries, such as polydimethylsiloxane-*block*-poly-caprolactone (PDMS-*b*-PCL), enables the design of a broad colour gamut. Interestingly, we find reversible and nonlinear thermochromism in both PDMS-*b*-PCL BBCPs and PDMS-*b*-PEO, but not in polystyrene-*block*-polycaprolactone (PS-*b*-PCL). This behaviour is caused by the interplay between a crystalline block and a soft one with a low glass transition temperature. This work establishes a new paradigm for inverse design in polymers, which allows predictive control of structural colours and thermochromism for display, dynamic photonic, sensing, and camouflage applications.

## Results

### Colour design model

To avoid the conventional trial-and-error method, we aim to develop an inverse design approach to make structural colours in BBCPs. By inputting a target colour into an "inverse design solver", we seek to directly obtain necessary side chain lengths ($n_{s,A}$, $n_{s,B}$), and backbone lengths ($n_{b,A}$, $n_{b,B}$), for given monomer chemistries (Fig. 1). The predicted structure is then synthesised and assembled to achieve the target colour, eliminating trial-and-error synthesis.

To construct the inverse design solver, we integrate a polymer physics model with an optical model to establish a direct link between molecular structures and macroscopic colours. The polymer physics model quantitatively maps molecular structures to nanostructures, while the optical model relates nanostructures to specific colours. The two models are coupled through a "nexus" parameter, domain spacing ($d$). Considering the multilayer structure is most common in bulk BBCPs, we adopt a multilayer optical model to obtain the best-fit spectrum from a target colour and then extract key parameters, namely domain spacing and refractive index. For given monomer chemistries, the refractive index is fixed, making the domain spacing the primary determinant of colour.

The theoretical domain spacings ($d_{\text{thy}}$) can be predicted from BBCP molecular structures using SS-SCF theory. Inspired by the framework of Zhulina et al. [33], we incorporate chemical disparities between backbone and side chains, which influence bottlebrush polymer conformation[35]. As shown in Fig. 1, for a given monomer chemistry, we first calculate molecular parameters including Kuhn length ($b$), monomer volume ($v$), and monomer contour length ($l$) for both backbone and side chains. We then define a characteristic length scale, $\widetilde{L}_A$, which is a function of two measurable parameters, critical crowding parameters ($\Phi_A^*$, $\Phi_B^*$) and interfacial tension ($\gamma_{A/B}$) (details in Supplementary Section 1.1–1.3). For crystalisable side chains, we

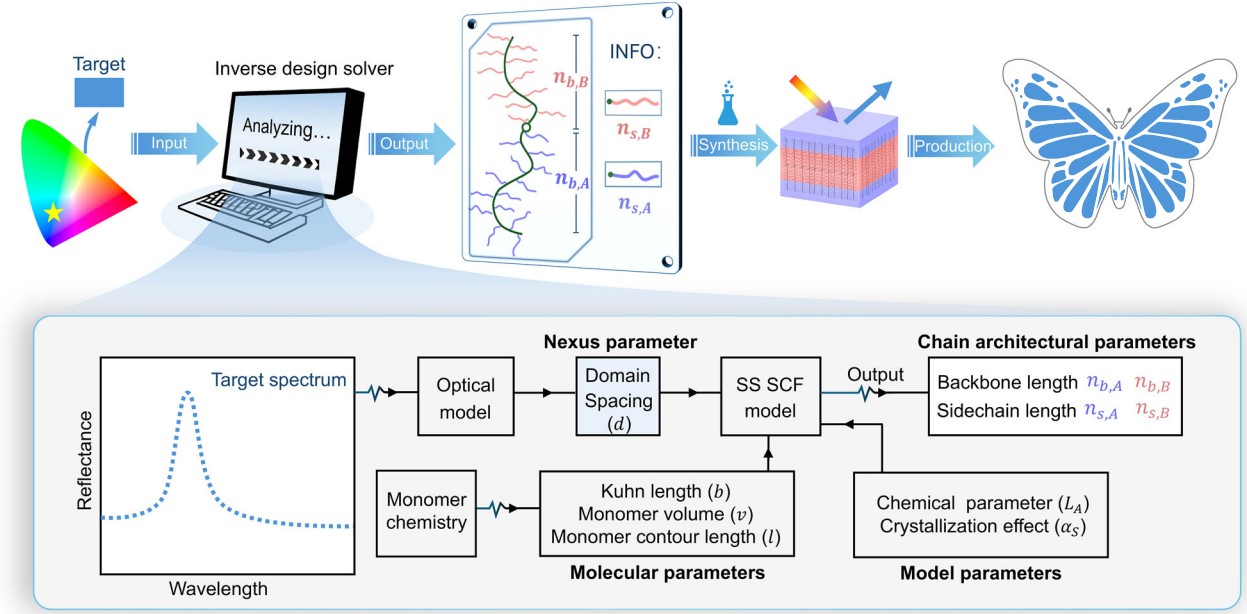

**Fig. 1 | Inverse design framework for structural colour production using BBCPs.** A target colour is input to an "inverse design solver" to obtain the required BBCP chain architectures for given monomer chemistries. The solver is a colour design model that integrates a multilayer optical model (relating spectra to domain sizes) and an SS-SCF model (bridging domain sizes to chain architectures) through a "nexus" parameter, namely domain spacing ($d$).

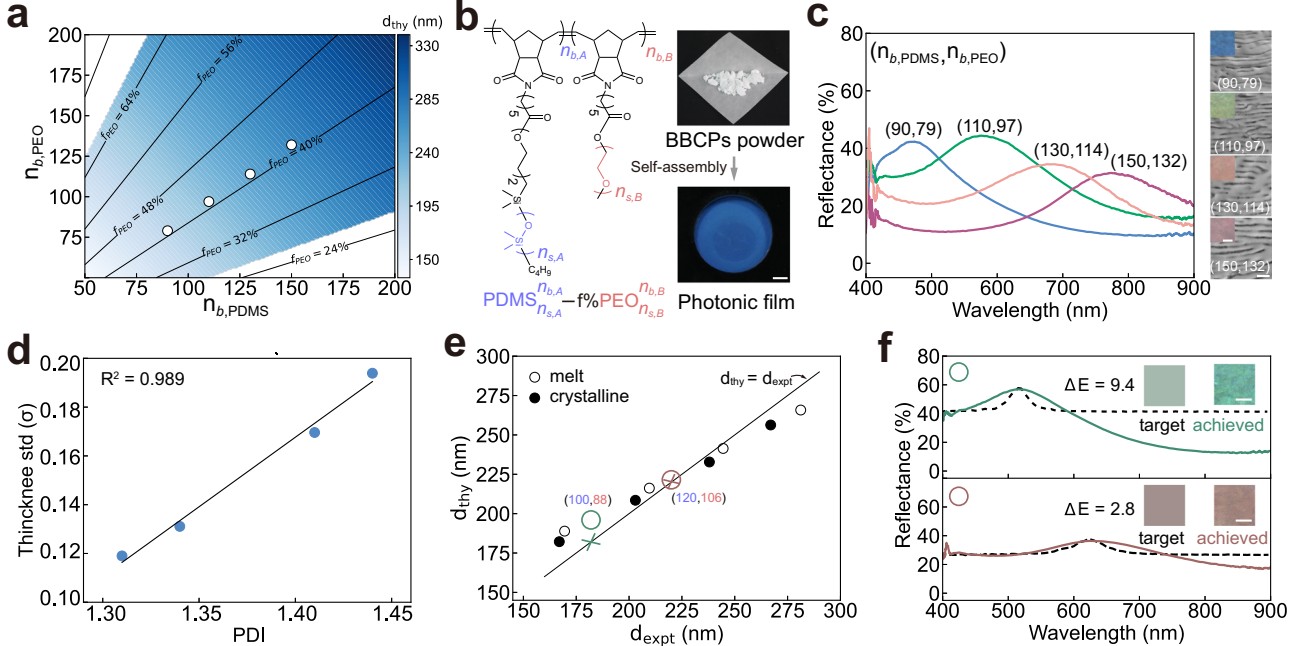

**Fig. 2 | Validation of the colour design model using PDMS-*b*-PEO BBCPs.**
**a** Predicted domain spacings as a function of backbone lengths for fixed side chain lengths ($n_{s, PDMS}$ = 68, $n_{s, PEO}$ = 105) based on the SS-SCF model in the absence of crystallisation. The blue region represents the lamellar phase (*L*) and the white region represents other structures such as cylindrical (*C, C'*) or spherical (*S, S'*) phases. **b** Molecular structure of PDMS-*b*-PEO, along with images of the as-synthesised material and photonic film after self-assembly. Scale bar, 5 mm.
**c** Reflectance spectra of photonic films with increasing molecular weights obtained by a microspectrometer. Insets are optical images and cross-sectional SEM images.

Scale bars are 50 $\mu m$ in optical images and 500 nm in SEM images. **d** Correlation between the optimal standard deviation of the layer thickness and BBCP polydispersity. **e** Comparison of theoretical and experimental domain spacings for four PDMS-*b*-PEO BBCPs at both melt (open circles) and crystallised (filled circles) states. Coloured crosses and circles are target and experimentally achieved domain spacings. **f** Reflectance spectra of target colours (dotted line) and corresponding experimentally achieved colours (solid line) using PDMS-*b*-PEO BBCPs with $n_{b, PDMS}$ = 100 and $n_{b, PEO}$ = 88, $n_{b, PDMS}$ = 120 and $n_{b, PEO}$ = 106.

assume that crystallisation influences domain spacing through increasing domain density and inducing side chain stiffening, phenomenologically described by a side chain stiffening parameter, $\alpha_s$ (details in Supplementary Section 1.3). Using both molecular and model parameters ($\tilde{L}_A$ and $\alpha_s$), the domain spacing is computed following Eq. (26) in Supplementary Information.

We validate the SS-SCF model by comparing the predicted theoretical domain spacings ($d_{thy}$) with the experimental domain spacings ($d_{expt}$). We choose PDMS-*b*-PEO as a model system due to its strong microphase separation, driven by a large Flory-Huggins parameter ($\chi \approx 0.21$)[36], low glass transition temperature of PDMS ($T_g$ ~ −125 °C)[37], and the crystallizability of PEO[38]. We expect PDMS and PEO blocks to undergo strong segregation to produce sharp interfaces and uniform self-assembly structures, thereby ensuring that real samples closely resemble those in the multilayer optical model. Using reported values for critical crowding parameter and interfacial tension of PDMS-*b*-PEO[39], we calculate $\tilde{L}_{PDMS}$ = 0.44 nm (details in Supplementary Section 1.3). This allows us to use the SS-SCF model to determine the range of chain architectural parameters required to obtain lamellar structures with domain spacings of 150–300 nm at the melt state (without crystallisation) (Fig. 2a). Within this range, we synthesise four PDMS-*b*-PEO BBCPs via sequential ring opening metathesis polymerisation (ROMP)[40], with different backbone degrees of polymerisation (Supplementary Figs. 3–8). BBCPs are assembled into bulk photonic films (>100 $\mu m$) by solution casting in a toluene atmosphere, followed by vacuum annealing at 100 °C for 8 h. (Fig. 2b, right). The toluene is used because it is a good solvent for most polymers, and its moderate evaporation rate offers enough time for polymer chains to undergo microphase separation to ordered structures during solution casting. Increasing block lengths ($n_{b, PDMS} + n_{b, PEO}$), causes a blue-to-red colour shift, consistent with the expected redshift (Fig. 2c).

From the measured reflectance spectra, we use a modified multilayer optical model to extract experimental domain spacing ($d_{expt}$). Cross-sectional SEM images reveal multilayer structures with some variations in layer thickness and orientations (insets in Fig. 2c). To account for these variations, we model the PDMS and PEO layer thickness with a truncated normal distribution and incident angles ranging from 0° to 24° (details in Supplementary Section 2). The means and standard deviations of the layer thickness distributions are determined through two steps of Bayesian optimisation, minimising colour differences between simulated and experimental spectra (Supplementary Figs. 2, 9). As shown in Fig. 2d, the standard deviation of layer thickness correlates linearly with the dispersity index (PDI) of PDMS-*b*-PEO. This suggests that this modified optical model can predict the thickness variations for each layer simply based on BBCP's PDI.

These samples crystallise at room temperature, as confirmed by differential scanning calorimetry (DSC) spectra (Supplementary Fig. 10). To first validate the SS-SCF model in the absence of crystallisation, we measure the reflectance spectra at the melt state ($T$ = 80 °C > $T_m$ = 57 °C, Supplementary Fig. 11) and extract the experimental domain spacing using the modified multilayered model. By fitting the calculated domain spacings to the experimental values at the melt state using Eq. (26) (details in Supplementary Section 1.3), we determine $\tilde{L}_{PDMS}$ = 0.44 nm, in excellent agreement with the theoretically calculated $\tilde{L}_{PDMS}$ = 0.44 nm. Incorporating a stiffening parameter of $\alpha_s$ = 1.96 yields theoretical predictions that align well with experimental domain spacings across all four BBCPs at the crystallised state (Fig. 2e, Supplementary Fig. 12). With the SS-SCF model validated under both melt and crystallised conditions, we can now use it to predict chain architectural parameters for a given domain spacing.

By integrating the modified multilayer optical model and SS-SCF model, we obtain a colour design model for inverse design of

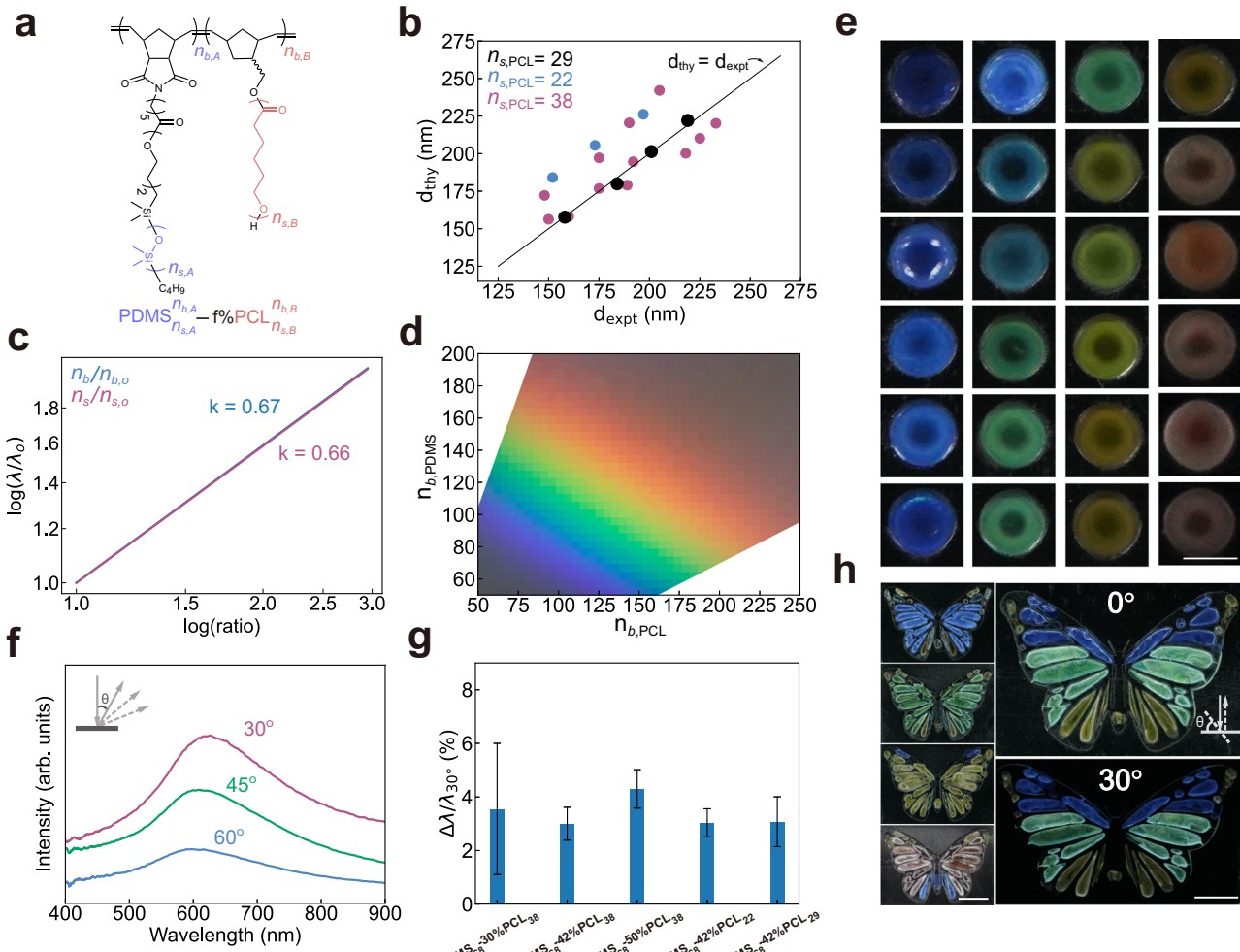

**Fig. 3 | Extension of the colour design model to PDMS-*b*-PCL BBCPs. a** Molecular structure of PDMS-*b*-PCL. **b** Comparison between theoretical and experimental domain spacings for PDMS-*b*-PCL BBCPs with different chain architectures (19 samples) at the crystallised state. Model parameters $\bar{L}_A$ and $\alpha_s$ are obtained by fitting four reference samples (black circles). **c** Effect of molecular architectural parameters on the reflectance peak with fixed side chain lengths ($n_{s,\text{PDMS}} = 68$, $n_{s,\text{PCL}} = 38$). **d** Predicted colour gamut from multilayer nanostructures when varying backbone lengths with fixed side chain lengths ($n_{s,\text{PDMS}} = 68$, $n_{s,\text{PCL}} = 38$). **e** Colour swatches produced by assembling BBCPs with various chain architectures. Scale bar, 1.5 mm. **f** Angle-resolved reflectance spectra of PDMS$_{68}^{150}$−50%PCL$_{38}^{210}$ film, measured with normal incident illumination and detection angles from 30° to 60°. **g** Reflectance peak shift ratios ($\Delta\lambda/\lambda_{30°}$) for five groups of BBCPs with different molecular parameters. $\Delta\lambda = \lambda_{30°} - \lambda_{45°}$ represents the peak shift when the detecting angle changes from 30° to 45° under fixed normal incident light. Data are presented as mean values ± standard errors of the mean (n = 3 for PDMS$_{68}$−30%PCL$_{38}$, PDMS$_{68}$−50%PCL$_{38}$, PDMS$_{68}$−42%PCL$_{22}$ and n = 4 for other samples). **h** Butterfly-shaped films displaying angle-dependent colours, guided by the colour design model. Scale bar, 5 mm.

structural colours. For a green colour with a reflectance peak at 520 nm, the required domain spacing is calculated as 195 nm by the optical model with determined average layer thickness and standard deviation of the thickness distribution. We then input the domain spacing to the SS-SCF model to determine the corresponding chain architectural parameters. Among the feasible solutions, we select a BBCP with $n_{s,\text{PDMS}} = 68$, $n_{s,\text{PEO}} = 105$, $n_{b,\text{PDMS}} = 100$, and $n_{b,\text{PEO}} = 88$. The synthesised BBCP (PDMS$_{68}^{100}$−46%PEO$_{105}^{88}$) assembles into a green film with a reflection wavelength of 516 nm, closely matching the target (Fig. 2f). Using the same side chain lengths, a red colour design yields a target domain spacing corresponding to $n_{b,\text{PDMS}} = 120$ and $n_{b,\text{PEO}} = 106$. The resulting film displays a red colour in agreement with the design, though with a slightly broadened spectrum.

To extend the model to BBCPs with different monomer chemistries, we choose PDMS-*b*-PCL due to PCL's crystallinity, comparable to PEO (Fig. 3a). Since the colour design model is currently limited to 1D lamellar structures, we need to determine the boundary conditions for PDMS-*b*-PCL BBCPs to form lamellar structures. Taking PDMS-*b*-PCL with $n_{s,\text{PDMS}} = 68$, $n_{s,\text{PCL}} = 29$ for example, we can calculate that lamellar structures fall in the range of PCL volume fraction of 28–68% at crystalline state (Supplementary Fig. 13). Guided by this, we synthesise and assemble four PDMS-*b*-PCL BBCPs with fixed side chain lengths ($n_{s,\text{PDMS}} = 68$, $n_{s,\text{PCL}} = 29$) and varying backbone lengths ($n_{b,\text{PDMS}} = 90-150$, $n_{b,\text{PCL}} = 115 - 190$) (Supplementary Figs. 14, 15). The resulting films show colours from blue to yellow (Supplementary Fig. 16) after annealing under vacuum at 100 °C for 8 h. Extending the annealing time causes almost no additional colour change, indicating that the lamellar structure reaches a metastable state if it is not at equilibrium (Supplementary Fig. 17). In addition, photonic films cast from different solvents show similar colours and their spectra peak positions are close to each other (Supplementary Fig. 18). This suggests these solvents do not cause much difference in colours of PDMS-*b*-PCL films, in contrast to a recent report that solvent can largely modulate the morphology and colour of polystyrene-*block*- polylactic acid (PS-*b*-PLA) BBCP films[41]. This is likely because PDMS and PCL side chains have fast chain dynamics compared to PS and PLA side chains and kinetically trapped structures during solvent evaporation can be effectively erased after subsequent thermal annealing. The

reproducibility of our experiments establishes the basis for reliable model-guided inverse design.

To determine $\widetilde{L}_A$ and $\alpha_s$ for PDMS-*b*-PCL system, we run two steps of optimisation in the same manner as for the PDMS-*b*-PEO system. According to Eq. (26) in Supplementary Information, we first run a linear fitting using four measured layer spacings $d$ in the melt state (obtained from reflection spectra in Supplementary Fig. 19a), combined with the polymer parameters listed in Supplementary Table 1. The slope gives $\widetilde{L}_A = 0.35$ nm, which is then used to optimise $\alpha_s$ based on layer spacing data in the crystallised state and gives $\alpha_s = 1.62$ (black circles in Fig. 3b, Supplementary Fig. 19). This two-step optimisation approach ensures that each parameter retains a clear physical meaning while minimising the coupling between parameters.

To evaluate the model's robustness across a broader design space, we then synthesise and assemble 22 more PDMS-*b*-PCL BBCPs with different side chains and volume fractions. The molecular weights range from $0.82 \times 10^6$ to $3.28 \times 10^6$ g/mol with PDI of 1.17–1.54 (Supplementary Fig. 20). All samples show lamellar structures, consistent to the prediction by the SS-SCF model (Supplementary Figs. 21–24 and Section 1.4). Only a few SEM images show lamellar structures mixed with local defects that are difficult to avoid for bulk films with thicknesses exceeding 100 μm. Experimental domain spacings, obtained from their reflectance spectra using the multilayer optical model, align well with the predicted spacings with these established $\widetilde{L}_A = 0.35$ nm and $\alpha_s = 1.62$ (Fig. 3b). We observe that predicted domain spacings are consistently larger than measured domain spacings for two sample groups (PDMS$_{68}$−50%PCL$_{38}$ and PDMS$_{68}$−42%PCL$_{22}$), likely due to variation in crystallinity of different PCL side chain lengths (Supplementary Fig. 25). This deviation may also arise from experimental variations between different batches such as uncertainty in measuring small volumes of Grubb's catalysts during synthesis. Therefore, these results support the model's applicability across a broad parameter space, including backbone lengths, side chain lengths, and volume fractions.

We further use the colour design model to visualise how the chain architecture affects the macroscopic colours. As shown in Fig. 3c, the reflectance peak ($\lambda$), which scales with domain spacing, follows a power-law dependence on both backbone and size chain lengths: $\lambda \sim (n_{s,\text{PDMS}} + n_{s,\text{PCL}})^{0.67}$ and $\lambda \sim (n_{b,\text{PDMS}} + n_{b,\text{PCL}})^{0.66}$. This trend offers design guidance for tuning colours in BBCPs. More importantly, we can use the model to generate an experimentally achievable colour space. For instance, by fixing $n_{s,\text{PDMS}} = 68$, $n_{s,\text{PCL}} = 38$, we can predict a full colour gamut by varying backbone lengths (Fig. 3d).

More BBCPs are synthesised and assembled to generate a full spectrum of colours (Fig. 3e, Supplementary Fig. 26). To demonstrate spatial patterning and practical applicability, four PDMS-*b*-PCL BBCPs are used to fabricate a butterfly-shaped coloured pattern (Fig. 3h). The resulting colours remain largely invariant with viewing angle, akin to the angle-independent behaviour of photonic glasses. Angle-resolved spectra confirm that colour changes are subtle with varying angles (Fig. 3f), with peak shifts as low as 3–4% across all samples with different chain architectures (Fig. 3g, Supplementary Fig. 27). This consistency suggests a uniform degree of structural disorder in these multilayer nanostructures, which contributes to their angular robustness.

### Crystallisation-induced thermochromism

Beyond the static colour control, we next investigate the temperature-dependent optical response in these BBCPs films. As shown in Fig. 4a, a representative PDMS-*b*-PCL film turns from orange to brown as the temperature decreases from 60 °C to 0 °C, with a full colour recovery upon heating. The temperature-responsive colour change is reversible after at least 25 heating and cooling cycles (Fig. 4b). This thermochromic behaviour is also observed in 17 PDMS-*b*-PCL photonic films with other chain architectures (Supplementary Fig. 28), demonstrating the generality of this thermochromism. We use ultra-small angle X-ray

scattering (USAXS) to examine structural changes in a representative photonic film after heating from 20 °C to 60 °C (Supplementary Fig. 29). In both states, we observe two scattering peaks with a *q*-value ratio of 1:2, suggesting the lamellar morphology retains after heating. However, the primary scattering peak shifts to lower $q$ values, corresponding to an increase in the lamellar spacing ($d$) from 145.5 nm to 156.3 nm. These values are in excellent agreement with the values calculated from optical spectra based on Bragg's law.

To systematically explore the kinetics of colour change, we combine optical spectroscopy, polarised optical microscopy (POM), and DSC. In-situ reflectance spectra show a sharp spectral shift around 17 °C during cooling and around 43 °C during heating (Supplementary Fig. 30). The colour remains nearly unchanged after the temperature increases from 60 °C to 100 °C (Supplementary Fig. 31), suggesting a narrow thermochromic window. Plotting the peak shift with temperature reveals an asymmetric, nonlinear response during both thermal cycles (Fig. 4c). POM intensity plots show sharp transitions at melting (41 °C) and crystallisation (19 °C), closely matching the observed colour change (Supplementary Fig. 32). DSC measurements further confirm the melting ($T_m = 48$ °C) and crystallisation ($T_c = 11$ °C) of PCL side chains, consistent with colour change. These results collectively demonstrate that the colour change in photonic films originates from the phase transition in crystalline PCL blocks.

The colour transition temperature depends on the PCL side chain length. As $n_{s,\text{PCL}}$ increases from 22 to 29 and 38, the colour change temperature rises from 13 °C to 19 °C during cooling and from 35 °C to 45 °C during heating (Fig. 4d, e). This trend aligns with DSC results of both homo bottlebrush PCL and PDMS-*b*-PCL with varying PCL chain lengths (Supplementary Figs. 33, 34). The elevated transition temperature observed in longer PCL chain lengths likely arises from stronger intermolecular interactions and the formation of extensive crystalline domains, which require higher thermal energy to melt. These findings show that we can tailor the colour-changing temperature simply by adjusting the PCL side chain lengths.

To quantitatively reveal the leading mechanism behind colour changes, we track the temperature-dependent evolution of film thickness and refractive index using a spectroscopic ellipsometer. As the temperature increases from 0 °C to 60 °C, the film thickness expands by 7.6%, while the refractive index at 589 nm decreases by 2.5% from 1.472 to 1.436 (Fig. 5a, b, Supplementary Fig. 35). Upon cooling, both the thickness and the refractive index recover with similar hysteresis to the melting and crystallisation transition of semicrystalline polymers. Using the measured temperature-dependent film thickness and refractive index, we apply Bragg's law to calculate the shift in reflectance peak positions (Eqs. 35–37 in Supplementary Section 4). The calculated peak shift not only captures the trend observed in experimental spectra, but also quantitatively agrees with the measured shift: a predicted change of 5.0% matches the measured 5.6% from 60 °C to 0 °C (Supplementary Fig. 36). This consistency extends across PDMS-*b*-PCL samples with different molecular architectures (Supplementary Fig. 37).

The PCL layer is expected to expand or shrink significantly more than the PDMS layer during thermal cycles due to its melting and crystallisation (Fig. 5c). To validate this, we further use the modified multilayer model to extract the layer thickness at both melt and crystallised states. Using the refractive indices of PCL and PDMS from ellipsometry measurements (details in Supplementary Section 4)[42], we find that the simulated and experimental spectra align when the PCL layer increases by 15.9% and the PDMS layer increases by 0.9% upon heating from 0 °C to 60 °C (Fig. 5d, e). This suggests that the PCL thickness variation is the dominant contributor to the colour change. Interestingly, the PCL homo bottlebrush film expands only by 5.5% in thickness, almost three times less than the thickness expansion of the PCL layer in the PDMS-*b*-PCL film after melting (Supplementary Fig. 38). This demonstrates that PCL crystallisation causes both volume

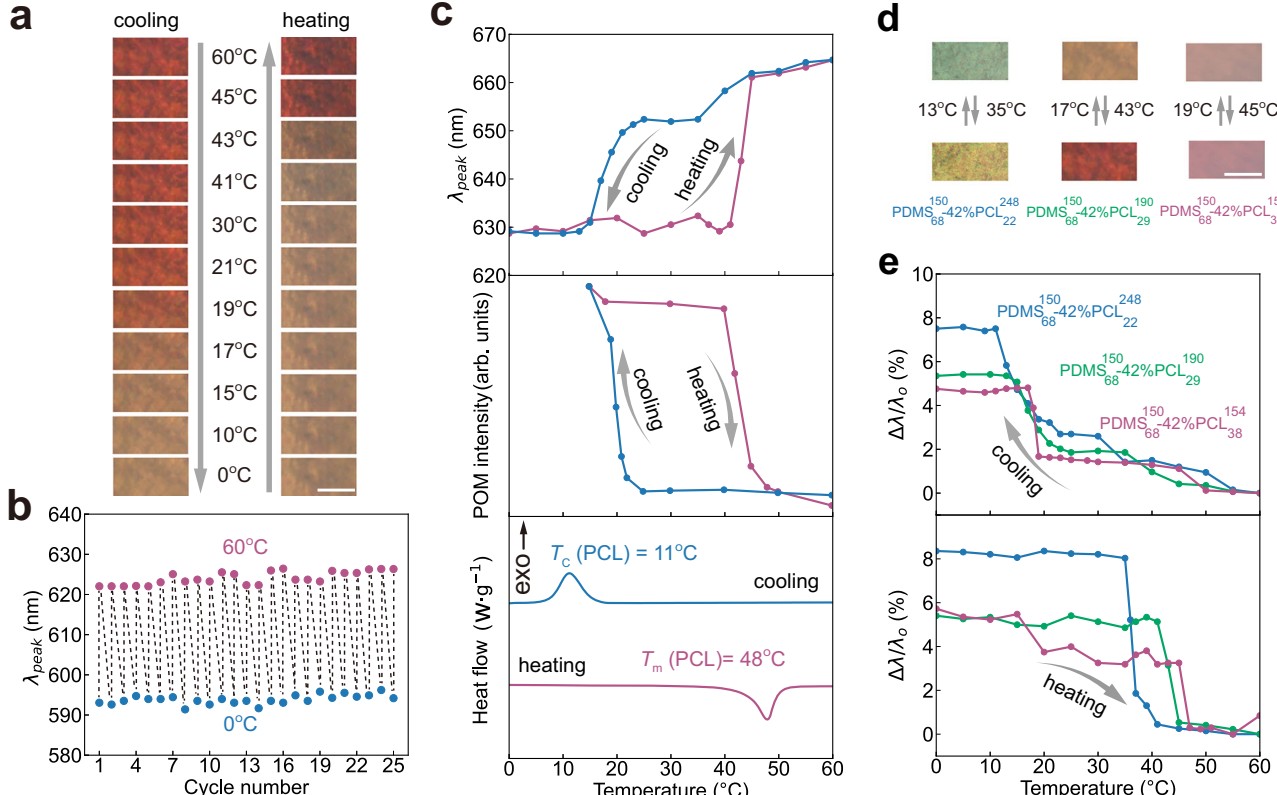

**Fig. 4 | Reversible, nonlinear thermochromism in PDMS-*b*-PCL photonic films.**
**a** Optical images of a PDMS-*b*-PCL film showing a reversible colour change between
0 °C and 60 °C during heating and cooling. Scale bar, 100 µm. **b** Reflectance peak
positions over 25 heating-cooling cycles, measured after equilibration at 0 °C and
60 °C. **c** Temperature-dependent evolution of reflectance peak wavelength,
intensity of POM images, and normalised heat flow. The sample in (**a**, **c**) is produced
by assembling PDMS$_{68}^{150}$−42%PCL$_{29}^{190}$ and the sample in (**b**) is produced by
assembling PDMS$_{68}^{130}$−42%PCL$_{29}^{165}$. **d** Optical images of three PDMS-*b*-PCL samples at
0 °C and 60 °C with transition temperatures indicated. Scale bar, 100 µm.
**e** Percentage of reflectance peak shift during heating and cooling for three BBCPs:
PDMS$_{68}^{150}$−42%PCL$_{38}^{154}$ (red curve), PDMS$_{68}^{150}$−42%PCL$_{29}^{190}$ (green curve), and
PDMS$_{68}^{150}$−42%PCL$_{22}^{248}$ (blue curve). All thermal scans are performed at a rate of
10 °C min$^{-1}$.

shrinkage and side chain stiffening, captured by the stiffening para-
meter ($\alpha_s$) in the SS-SCF model.

We examine other semicrystalline BBCPs to assess the universality
of thermochromism. A PDMS-*b*-PEO film exhibits a colour shift upon
heating to 80 °C, exceeding the PEO's melting temperature
($T_m$ = 57 °C) (Fig. 5f). In contrast, a coloured PS-*b*-PCL film shows
negligible colour change even at 120 °C (Fig. 5g, chemical character-
isations in Supplementary Figs. 7, 39). DSC analysis shows that the high
glass transition temperature of PS ($T_g$ = 102 °C) restricts PCL
crystallisation[43], resulting in minimal colour change (Supplementary
Fig. 40). PDMS-*b*-PCL and PDMS-*b*-PEO show significant crystallisation,
likely facilitated by the low-$T_g$ PDMS block. These results demonstrate
that BBCPs comprising a crystalline block with a soft, low-$T_g$ block
enable temperature-responsive photonic films. Leveraging this ther-
mochromic property, we design a chameleon-inspired pattern with a
temperature-sensitive PDMS-*b*-PCL body and a non-responsive PS-*b*-
PCL tree trunk. When heating from 20 °C to 60 °C, the chameleon's
blue and yellow stripes turn to green and orange, while the tree trunk
remains unchanged. This suggests different BBCPs can be selected to
achieve spatially tunable thermochromic patterning.

## Discussion
In summary, we develop an inverse design approach for structurally
coloured BBCPs by establishing a quantitative colour design model that
integrates an SS-SCF model with a modified multilayer optical model,
quantitatively linking chain architectures to macroscopic colours. The

colour design model is experimentally validated through using PDMS-*b*-
PEO with varying chain architectures, and can be extended to other
BBCPs chemistries, such as PDMS-*b*-PCL, to make a full spectrum of
colours. The inverse design model is currently limited to lamellar pho-
tonic structures and can be potentially extended to other 2D or 3D
nanostructures in the future. This will require a more accurate SS-SCF
model to predict 2D and 3D photonic structures with domain spacing as
large as hundreds of nanometres, along with experimentally producing
large-scale non-lamellar BBCP structures.

Notably, reversible, nonlinear thermochromic behaviour is
observed in both PDMS-*b*-PCL and PDMS-*b*-PEO, but not in PS-*b*-PCL.
The unique thermochromism arises from the interplay between a
crystalline block and a soft, low-$T_g$ block, as confirmed by dynamic
optical measurements and multilayer optical modelling incorporat-
ing in-situ refractive index and thickness data. This work provides an
inverse design approach that enables precise structural colour
engineering in polymer materials but also offers fundamental
insights into the thermochromic behaviour of semicrystalline BBCPs,
paving the way for the design of advanced responsive photonic
materials.

## Methods
A complete set of detailed synthesis procedures for small organic
molecules and polymers, scanning electron microscopic images,
and reflectance spectral data are available in the Supplementary
Information.

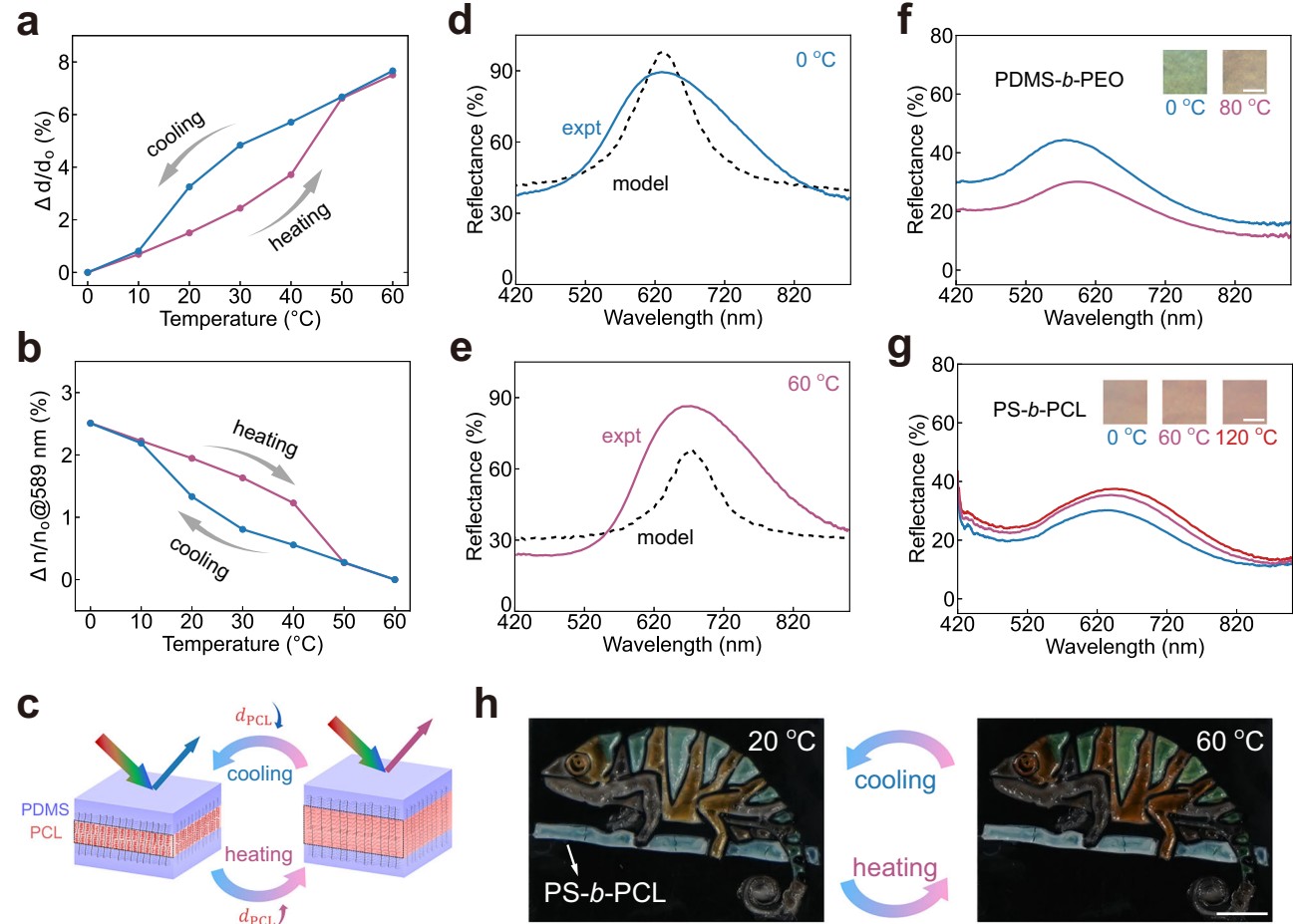

**Fig. 5 | Investigation of the thermochromic mechanism in BBCP photonic films.**
**a**, **b** Temperature-dependent changes in (**a**) refractive index and (**b**) thickness during heating and cooling cycles. Each data point is measured within 20 s at the corresponding temperature. The sample is assembled from $PDMS_{68}^{150}-42\%PCL_{29}^{190}$. **c** Schematic illustration showing domain spacing variations during thermal cycles. **d**, **e** Comparisons between experimental reflectance spectra taken by a microspectrometer with optical model predictions at 0 °C and 60 °C. **f**, **g** Reflectance spectra taken by a microspectrometer and optical images at different temperatures for (**f**) $PDMS_{68}^{110}-46\%PEO_{105}^{97}$ and (**g**) $PS_{37}^{110}-40\%PCL_{22}^{113}$ films. Scale bars: 50 µm. **h** Chameleon-shaped pattern composed of thermochromic PDMS-*b*-PCL regions (body) with non-responsive PS-*b*-PCL (tree trunk), demonstrating selective colour change upon heating and cooling. Scale bar, 5 mm.

## Materials and instrumentation
All chemicals were used as received from Aladdin Chemicals, Adamas-beta or Sigma-Aldrich, unless otherwise noted. Proton nuclear magnetic resonance (¹H NMR) spectra were recorded on Bruker AVANCE III-400 spectrometers, with chemical shifts (ppm) referenced to $CDCl_3$ (7.26 ppm). Gel permeation chromatography (GPC) was performed on Waters Alliance e2695 instrumentation containing two Styragel HR 5E $300 \times 7.8\ mm^2$ columns connected in series with a DAWN HELEOS multi-angle laser light scattering (MALS) detector and a refractive index (RI) detector. Tetrahydrofuran (THF) was used as the eluent with a flow rate of 1.0 mL min⁻¹ at 35 °C. Absolute molecular weights were determined from the MALS detector signals and $dn/dc$ values measured for each injection.

## General ROMP procedures for the BBCP syntheses
BBCPs were synthesised via sequential ROMP in a glovebox (Etelux Lab2000) with chemical reactions illustrated in Supplementary Fig 4. The first norbornene-based macromonomer was dissolved in anhydrous THF under stirring in a glass vial. The G3 catalyst was then prepared by dissolving it in anhydrous THF at a concentration of 2 mg mL⁻¹. Under rapid stirring, a calculated volume of this catalyst solution was quickly added to the macromonomer solution to achieve the desired degree of polymerisation. After stirring for

10 min, a second norbornene-based macromonomer was introduced, and the reaction continued for 4 h. The mixture was then removed from the glovebox and quenched with excess ethyl vinyl ether. Purification was achieved by precipitating the product into methanol, yielding the BBCP powders. The samples were characterised by GPC and ¹H NMR spectroscopy. GPC traces of BBCPs and their corresponding macromolecules were provided in Supplementary Figs 8, 14, 20.

## Preparation of photonic films
BBCP solutions were first prepared by dissolving 50 mg of BBCPs in 500 µL of toluene at 60 °C. The solutions were then drop-cast into a customised glass well and allowed to dry in a toluene atmosphere. After complete solvent evaporation, the sample was annealed in a vacuum oven at 100 °C for 8 h. The films showed various colours depending on the BBCPs' molecular architectures.

## SEM measurement
Scanning electron microscope (SEM) was performed on a Zeiss Gemini 300 instrument at 3 kV with a working distance of 8–10 mm. BBCP films were immersed in liquid nitrogen for 15 min and cut to make cross-sectional samples. The cross-sectional surfaces were then gold-coated using a sputter coater (Quorum SC7620) for SEM imaging.

## USAXS measurement

USAXS was performed on Beamline BL10U1 at Shanghai Synchrotron Radiation Facility, China. The wavelength of the X-ray was 1.24 Å, and the sample-to-detector distance was 27600 mm. The detector (Pilatus 2 M) had a resolution of 1475 × 1679 pixels and a pixel size of $75\,\mu m$. Data analysis was performed using the Fit2D software. BBCP solutions were drop-cast onto a Kapton tape and allowed to dry in a toluene atmosphere. The samples were subjected to thermal annealing under vacuum at 100 °C for 8 h before USAXS measurements.

## Reflectance spectrum measurement

We had three different optical setups to measure the reflectance spectrum of BBCP films. To characterise large-area BBCP films, we employed an integrating sphere (Ideaoptics IS20, sphere size 20 mm) equipped with a deuterium-halide light source (Ideaoptics iDH2000-BSC) and an optical-fibre-based spectrometer (Ideaoptics PG2000-PRO-EX, 195-1123 nm, slit width 25 μm) in conjunction with a PTFE white board. We took dynamic spectra and microscopic images during heating-cooling cycles using an optical microscope coupled with a thermal stage (PE-35130, Suzhou Keyiqian Precision Equipment Co., Ltd.) and a fibre-optic spectrometer (Ideaoptics PG2000-PRO-EX). Considering there were temperature differences between the upper and lower surfaces, we attached a thermocouple to the sample surface and recorded it as the in-situ temperature. Reflectance spectra were obtained using a PTFE diffuse reflector sheet (Thorlabs, PMR10P1) as a white reference. The reflectance spectra were smoothed using a rolling window method with a window size of 10. To quantify the angle dependence, a custom-built goniometer was employed to measure the angle-resolved reflectance by independently varying the incident and detection angles. Macroscopic and microscopic images were captured using a digital camera (Sony Direct-PmE1) and an optical microscope (Nikon, LVD100), respectively.

## Differential scanning calorimetry measurement

The thermal behaviour of BBCPs was characterised using a differential scanning calorimeter (Q200, TA Instruments). To eliminate thermal history, samples were first heated from ambient temperature to 150 °C at a rate of 10 °C min$^{-1}$, maintained isothermally for two minutes, then cooled to −70 °C at 10 °C min$^{-1}$ with a two-minute stabilisation period. A subsequent heating cycle to 150 °C at 10 °C min$^{-1}$ was performed under a nitrogen atmosphere. The crystallisation temperature ($T_c$) was determined from the exothermic peak during the cooling scan, while the melting temperature ($T_m$) and crystallinity degree ($X_c$) were derived from the endothermic peak integration in the second heating scan.

## Data availability

The authors declare that the main data supporting the findings of this study are available within the article and its Supplementary Information files. Raw data have been deposited in the Zenodo databased with https://doi.org/10.5281/zenodo.17382811. Data are available from the corresponding author upon request.

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

## Acknowledgements

M. Xiao thanks the financial support by the National Natural Science Foundation of China (52203273), State Key Laboratory of Advanced Polymer Materials (Grant No. sklpme2023-2-14), Beamtime Key Project at the Shanghai Synchrotron Radiation Facility (2024 SSRF ZD 508355), and the Fundamental Research Funds for the Central Universities. The authors thank Hao Zhang and Feng Tian (Shanghai Advanced Research Institute, Chinese Academy of Sciences) for their technical assistance with USAXS measurements. The authors also thank Zhan Li for his help in BBCPs synthesis and Qiang Fu for using the glove box in his lab.

## Author contributions

M. Xiao and D. Yang conceived the project and designed the experiments. D. Yang, C. Zhang, and Q. Li performed the experiments and ran optical modelling. H. Liang performed SS-SCF calculations. P. Shao and R. Wen performed the refractive index measurement and analysis. M. Xiao, D. Yang, H. Liang, L. Jiang, and C. Zeng analyzed the whole data and interpreted the experiments. D. Yang and M. Xiao wrote the paper with inputs from all authors. M. Xiao, L. Jiang, Y. Dan, and Y. Huang supervised the project and acquired the funding.

## Competing interests

M. Xiao, D. Yang, C. Zhang, and L. Jiang are inventors on a pending Chinese patent application (Application No. 202511513123X, filed on 22 October 2025) related to the inverse design of structural colours in BBCPs, and they declare no other competing interests. The remaining authors declare no competing interests.
