## [Transparent Peer Review file · Nature Communications]

Inverse design of structural colours in polymeric films with crystallization-induced reversible thermochromism

Corresponding Author: Professor Ming Xiao

Version 0:

Reviewer comments:

Reviewer #1

(Remarks to the Author)

This study establishes a quantitative model linking molecular structure, nanostructure, and optical properties, enabling the inverse design of structural colors in brush-like polymers. Furthermore, it reveals a novel mechanism for thermochromism induced by the combination of 'crystalline blocks and low-Tg soft segments,' providing an effective route to advanced smart photonic materials. In my opinion, this work represents a valuable addition at the interface of polymer science and photonics. The authors provide solid data to well support their conclusions, and the manuscript is well written for a clear presentation. Therefore, I would like to recommend the publication of the work in Nature Communications after addressing the following minor points:

1> Why did the authors make the samples using toluene as the solvent? Is the lamellar morphology approaching an equilibrium state after annealing at 100 °C for 8 hours?

2> The inset SEM images in Figure 2c may be moved outside the graph and enlarged to show clearly the lamellar structures.

3> Based on the DSC curve, the authors may need to calculate the crystallinity of the PCL block.

4> Supplementary Fig. 7 notes minor color differences between simulated and experimental spectra. The manuscript attributes this to "assumptions of smooth layers without defects", but deeper investigation is needed. Could polymer polydispersity (e.g., PDI in Table S3) or interfacial roughness cause these deviations?

5> In the "Synthesis of PS macromonomers section", we would like to verify whether the molar amount of initiator added may contain an error. Based on the molar equivalents reported, it appears that the stated initiator loading would not yield macromonomers with the degree of polymerization described in the subsequent text.

6> Regarding the "direct initiation" method described in this research for preparing PS macromonomers, we acknowledge the inherent risk that bimolecular termination could unavoidably produce species functionalized with norbornenyl groups at both ends. This presents a significant potential to initiate cross-linking during the subsequent ROMP step. To address this concern and demonstrate the absence of significant multimodal dispersity arising from such side products, please provide the GPC traces for both the synthesized PS macromonomers and the resulting PS-containing BCCPs.

7> Can the inverse design approach be extended to other 2D or 3D photonic structures? In the conclusion, the authors may wish to make some comments on the limitation of the current approach and on how to improve further to make it applicable to diverse structures.

Reviewer #2

(Remarks to the Author)

The manuscript presents a model for predicting the self-assembly behavior and resulting optical properties of bottlebrush block copolymer systems. The work is timely and relevant to the development of polymer-based photonic materials. The methodology is clearly described, and the connection between theoretical predictions and experimental validation is compelling. However, I have several comments regarding the applicability and generalizability of the model.

1. The manuscript would benefit from a more explicit discussion of the limitations and boundary conditions of the proposed model. For example, it is not entirely clear under what ranges of polymer composition and architecture the model is expected to remain predictive. Clearer articulation of these boundaries would improve the generalizability and utility of the framework for broader applications.

2. The authors are encouraged to clarify the phase limitations of bottlebrush block copolymers in the context of their model. The optical model is based on the reflection from lamellar structures, and the subsequent molecular design appears to assume that lamellae will form under experimental conditions. However, some of the SEM images provided in the Supporting Information (e.g., Figures 16 and 18) display morphologies that may not be strictly lamellar. This raises the question of how the authors ensure that the designed polymers indeed self-assemble into lamellar phases necessary for the model's applicability.

3. The authors are encouraged to clarify the potential discrepancies between the SS-SCF model and the experimental results obtained from solution-processed films. The SS-SCF model was used to extract architectural parameters of PDMS-b-PEO bottlebrush block copolymers but does not account for solvent effects. In contrast, the experimental films were prepared via solution casting followed by solvent vapor annealing in a toluene atmosphere. Since the choice of solvent during SVA can significantly influence the self-assembly pathway and final morphology, it would be helpful for the authors to discuss how these processing conditions might affect the applicability of the model predictions to the experimentally observed structures and optical responses.

4. The authors may consider clarifying the angular range used in the simulations versus experiments. While the model incorporated angular scans from 0° to 24°, the reflectance spectra were acquired with an integrating sphere. This difference could affect the comparison and should be briefly addressed.

Reviewer #3

(Remarks to the Author)

[Editorial Note: See end of file]

Please refer to the attachment "631875_0_report.pdf".

Version 1:

Reviewer comments:

Reviewer #1

(Remarks to the Author)

The authors have addressed the reviewers' comments appropriately in the revised manuscript. The revisions are satisfactory, and the manuscript is now suitable for acceptance and publication.

Reviewer #2

(Remarks to the Author)

I would like to thank the authors for their detailed responses and revisions. The concerns I raised have been satisfactorily addressed, and I have no further concerns. I find the manuscript suitable for publication.

Reviewer #3

(Remarks to the Author)

I would like to thank the authors for addressing my comments and suggestions for change, in full. In my opinion the present form of the article is suitable for publication in Nat. Comm.

Response to Reviewer #1

Comment 1.1

This study establishes a quantitative model linking molecular structure, nanostructure, and optical properties, enabling the inverse design of structural colours in brush-like polymers. Furthermore, it reveals a novel mechanism for thermochromism induced by the combination of 'crystalline blocks and low- T_g soft segments,' providing an effective route to advanced smart photonic materials. In my opinion, this work represents a valuable addition at the interface of polymer science and photonics. The authors provide solid data to well support their conclusions, and the manuscript is well written for a clear presentation. Therefore, I would like to recommend the publication of the work in Nature Communications after addressing the following minor points:

Response 1.1

We thank the reviewer for the thorough review of our manuscript and the positive comments about the work.

Comment 1.2

Why did the authors make the samples using toluene as the solvent? Is the lamellar morphology approaching an equilibrium state after annealing at 100°C for 8 hours?

Response 1.2

We appreciate the reviewer's question. We use toluene because it is a good solvent for most polymers such as PDMS, PEO, PCL, and PS. In addition, its moderate evaporation rate offers enough time for polymer chains to undergo microphase separation and form lamellar structures during solution casting. To make it clear, we have made revision to the main text:

[Main Text, Lines 141-144 on Page 6] "The toluene is used because it is a good solvent for most polymers and its moderate evaporation rate offers enough time for polymer chains to undergo microphase separation to ordered structures during solution casting."

It is never a simple task to conclude whether the structures reach equilibrium, but we can use experiments to examine if it is a metastable state. We have added the reflection spectra of a representative sample after different annealing hours in new Supplementary Fig. 15. The results show that the spectra become stable after 8 hours of annealing, and further annealing leads to negligible colour change. This indirectly suggests that the sample likely has reached a metastable state. Accordingly, we have made revisions in the revise manuscript:

[Main Text, Lines 221-224 on Page 9] "The resulting films show colours from blue to yellow (Supplementary Fig. 14) after annealing under vacuum at 100°C for 8 hours. Extending the

annealing time causes almost no additional colour change, indicating that the lamellar structure reaches a metastable state if it is not at equilibrium (Supplementary Fig. 15).”

Supplementary Fig. 15. Reflectance spectra taken by an integrating sphere of PDMS¹¹⁰-42%PCL¹³⁹ films with different annealing time. The dashed line denotes the averaged reflectance spectrum of five measurements at different locations of one sample and the shaded area represents the standard deviation from five measurements.

Comment 1.3

The inset SEM images in Figure 2c may be moved outside the graph and enlarged to show clearly the lamellar structures.

Response 1.3

We thank the reviewer’s suggestions and have moved the inset SEM images outside the graph in Figure 2c to clearly display the lamellar structure, as shown below.

Fig. 1. c, Reflectance spectra of photonic films with increasing molecular weights obtained by microspectrometer. Insets are optical images and cross-sectional SEM images. Scale bars are 50 μm in optical images and 500 nm in SEM images.

Comment 1.4

Based on the DSC curve, the authors may need to calculate the crystallinity of the PCL block.

Response 1.4

Based on the reviewer's suggestions, we have summarized the degree of PCL crystallinity in Table S2. The crystallinity varies from 36% to 45% for PDMS-*b*-PCL BBCPs and we have annotated the corresponding values on the DSC curves (new Supplementary Figs 31 and 32).

Comment 1.5

Supplementary Fig. 7 notes minor colour differences between simulated and experimental spectra. The manuscript attributes this to "assumptions of smooth layers without defects", but deeper investigation is needed. Could polymer polydispersity (e.g., PDI in Table S3) or interfacial roughness cause these deviations?

Response 1.5

Yes, both polymer dispersity and interfacial roughness will contribute to the colour difference between simulated and experimental spectra. In the optical model, we assume each PDMS and PEO layer to be flat with fixed thickness, while cross-sectional SEM images of real samples show each layer is curved to some degree and thickness variation exists in the same layer. These discrepancies are likely caused from polymer polydispersity and interfacial roughness. Polymer polydispersity can broaden the distribution of lamellar thicknesses, which in turn smears the reflectance peaks. Moreover, the experimental films are bulk-cast (>100 μm thick) and inevitably contain interfacial roughness and local imperfections. Such roughness can cause light scattering and shift or broaden reflectance features. We have revised the Section S2 to clarify this point and acknowledge the potential role of polydispersity and roughness in the observed discrepancies:

[SI, Section S2 in Page 10] "However, some discrepancy remains between the calculated and measured spectra. We attribute this difference to the fact that the optical model assumes flat, smooth, and uniform defect-free layers, whereas real samples contain curved layers with interfacial roughness likely caused by polymer dispersity and self-assembly defects."

Comment 1.6

In the "Synthesis of PS macromonomers section", we would like to verify whether the molar amount of initiator added may contain an error. Based on the molar equivalents reported, it appears that the stated initiator loading would not yield macromonomers with the degree of polymerization described in the subsequent text.

Response 1.6

We thank the reviewer for the catching up the typo. We have corrected the molar amount of initiator from 9.4 to 0.94 in Section S3.

Comment 1.7

Regarding the "direct initiation" method described in this research for preparing PS macromonomers, we acknowledge the inherent risk that bimolecular termination could unavoidably produce species functionalized with norbornenyl groups at both ends. This presents a significant potential to initiate cross-linking during the subsequent ROMP step. To address this concern and demonstrate the absence of significant multimodal dispersity arising from such side products, please provide the GPC traces for both the synthesized PS macromonomers and the resulting PS-containing BBCPs.

Response 1.7

We agree that bimolecular termination during the "direct initiation" route could generate polystyrene species bearing norbornenyl groups at both ends, which might lead to undesired cross-linking upon ROMP. This termination seems not obvious in our samples, as shown in GPC traces of both the synthesized PS macromonomers and the corresponding PS-containing BBCPs (new Supplementary Fig. 37).

Supplementary Fig.37. GPC curves of NBI-PS₃₇ and PS₃₇¹¹⁰-40%PCL₂₂¹¹³.

Comment 1.8

Can the inverse design approach be extended to other 2D or 3D photonic structures? In the conclusion, the authors may wish to make some comments on the limitation of the current approach and on how to improve further to make it applicable to diverse structures.

Response 1.8

We thank the reviewer for the suggestion. The general concept of inverse design can be extended to other complex photonic structures, such as 2D hexagonal cylinder or 3D cubic phase. However, there are two main technical challenges. First, the current self-consistent field theory to predict 2D and 3D photonic structures deviates from experimental observation for domain spaces of ~ 100 nm. It requires further validation for larger domain spacing. Second, it is nontrivial to experimentally obtain 2D and 3D photonic structures due to high curvatures in these structures are not compatible with the rigid bottlebrush polymers. We have added a paragraph in the conclusion to acknowledge the current limitations and to outline possible directions for extending the method to diverse 2D and 3D photonic systems:

[Main Text, Lines 385-389 on Page 15-16] “The inverse design model is currently limited to lamellar photonic structures and can be potentially extended to other 2D or 3D nanostructures in the future. This will require a more accurate SS-SCF model to predict 2D and 3D photonic structures with domain spacing as large as hundreds of nanometers, along with experimentally producing large-scale non-lamellar BBCP structures.”

Response to Reviewer #2

Comment 2.1

The manuscript presents a model for predicting the self-assembly behavior and resulting optical properties of bottlebrush block copolymer systems. The work is timely and relevant to the development of polymer-based photonic materials. The methodology is clearly described, and the connection between theoretical predictions and experimental validation is compelling. However, I have several comments regarding the applicability and generalizability of the model.

Response 2.1

We thank the reviewer for accurately summarizing the work and positive comments.

Comment 2.2

The manuscript would benefit from a more explicit discussion of the limitations and boundary conditions of the proposed model. For example, it is not entirely clear under what ranges of polymer composition and architecture the model is expected to remain predictive. Clearer articulation of these boundaries would improve the generalizability and utility of the framework for broader applications.

Response 2.2

We thank the reviewer for the comment. The inverse design framework is based on the integration of an optical model and a polymer physics model by assuming the photonic structures are 1D multilayered. Under the framework of 1D lamellar structure, our big assumption is that χ between two blocks is high enough to generate strong segregation and sharp interfaces between two domains. The variation in χ will not only shifts the phase boundary of BCCPs, but also smear the interfaces and create nonuniform structures, leading to more inaccuracy in the multilayer optical model. We have discussed the limitation of colour design model in conclusion part as mentioned in Response 1.8. In addition, we have added relevant discussion in the section of color design model:

[Main Text, Lines 130-132 on Page 5] “We expect PDMS and PEO blocks to undergo strong segregation to produce sharp interfaces and uniform self-assembly structures, thereby ensuring that real samples closely resemble those in the multilayer optical model.”

To illustrate what ranges of polymer compositions and architectures are working within the colour design model, we can use our modified self-consistent mean field theory to delineate boundary conditions for producing lamellar structures given the chemistry of BCCPs. Taking PDMS-*b*-PCL with $n_{s,PCL} = 29$ for example, we can calculate that lamellar structures fall in the range of PCL volume fraction of 28-68% at the crystalline state, as shown in new Supplementary Fig. 11. Accordingly, we have made relevant edits to the revised manuscript as:

[Main Text, Lines 214-220 on Page 9] “Since the colour design model is currently limited to 1D lamellar structures, we need to determine the boundary conditions for PDMS-*b*-PCL BBCPs to form lamellar structures. Taking PDMS-*b*-PCL with $n_{s,PDMS} = 68, n_{s,PCL} = 29$ for example, we can calculate that lamellar structures fall in the range of PCL volume fraction of 28-68% at crystalline state (Supplementary Fig. 11). Guided by this, we synthesize and assemble four PDMS-*b*-PCL BBCPs with fixed side chain lengths ($n_{s,PDMS} = 68, n_{s,PCL} = 29$) and varying backbone lengths ($n_{b,PDMS} = 90-150, n_{b,PCL} = 115-190$).”

Supplementary Fig. 1. Theoretical phase diagram of PDMS-*b*-PCL with $n_{s,PDMS} = 68, n_{s,PCL} = 29$ at the crystallized state based on SS-SCF theory, including 1D lamellar (L), 2D cylindrical (C, C'), and 3D spherical phases (S, S'). The blue points mark the boundary volume fraction to form a lamellar phase, and the red point is the volume fraction we choose to make, which represent all BBCPs with various molecular weights at $f_{PCL}=42\%$.

Comment 2.3

The authors are encouraged to clarify the phase limitations of bottlebrush block copolymers in the context of their model. The optical model is based on the reflection from lamellar structures, and the subsequent molecular design appears to assume that lamellae will form under experimental conditions. However, some of the SEM images provided in the Supporting Information (e.g., Figures 16 and 18) display morphologies that may not be strictly lamellar. This raises the question of how the authors ensure that the designed polymers indeed self-assemble into lamellar phases necessary for the model's applicability.

Response 2.3

We thank the reviewer for raising this concern about the model's limitation. As mentioned in Response 1.8 and Response 2.2, we clarify our colour design is currently limited to 1D lamellar structures.

To validate our colour design model, we used the SS-SCF calculations to delineate the volume fraction boundaries for lamellae given the chemistry of the bottlebrush block copolymers. We then synthesized PDMS-*b*-PEO or PDMS-*b*-PCL BBCPs within this window. Experimentally, the majority of PDMS-*b*-PEO or PDMS-*b*-PCL samples assembled into lamellar phases, as confirmed by SEM. While only a few SEM images (e.g., new Supplementary Figs. 20 and 22) showed lamellar structures mixed with locally imperfect morphologies. That is likely because the films are bulk-cast materials with thicknesses exceeding 100 μm , where some defects are unavoidable. We have added new explanations in revised manuscript:

[Main Text, Lines 246-249 on Page 10] “All samples show lamellar structures, consistent to the prediction by the SS-SCF model (Supplementary Figs. 19-22 and Section S1.4). Only a few SEM images show lamellar structures mixed with local defects that are difficult to avoid for bulk films with thicknesses exceeding 100 μm .”

Comment 2.4

The authors are encouraged to clarify the potential discrepancies between the SS-SCF model and the experimental results obtained from solution-processed films. The SS-SCF model was used to extract architectural parameters of PDMS-*b*-PEO bottlebrush block copolymers but does not account for solvent effects. In contrast, the experimental films were prepared via solution casting followed by solvent vapor annealing in a toluene atmosphere. Since the choice of solvent during SVA can significantly influence the self-assembly pathway and final morphology, it would be helpful for the authors to discuss how these processing conditions might affect the applicability of the model predictions to the experimentally observed structures and optical responses.

Response 2.4

We thank the reviewer for raising the question about solvent effect. To examine the possible influence of solvent, we have prepared several representative films by solvent vapor annealing using chloroform, tetrahydrofuran, and toluene. The optical photographs show similar colours and the spectra peak positions are close to each (new Supplementary Fig. 16). The reflectance intensity varies with solvent likely due to different sample thicknesses. These solvents seem not cause much difference in colours of assembled PDMS-*b*-PCL films, in contrast to a recent report that solvent can largely modulate the morphology and colour of PS-*b*-PLA BBCP films (*Soft Matter*, 2025, 21, 2217). We interpolate that PDMS and PCL have fast chain dynamics due to their low glass transition temperatures and solvent-induced kinetically trapped structures can be effectively erased during the subsequent thermal annealing step. In addition, we also demonstrate that our samples undergo enough annealing time to a stable state as noted in Response 1.2. Therefore, it is appropriate to apply the SS-SCF model to these solid-state films

for predicting lamellar phases and guiding polymer design. We have added revisions to the main text as:

[Main Text, Lines 224-233 on Page 9] “In addition, photonic films cast from different solvents show similar colours and their spectra peak positions are close to each (Supplementary Fig. 16). This suggests these solvents do not cause much difference in colours of PDMS-*b*-PCL films, in contrast to a recent report that solvent can largely modulate the morphology and colour of PS-*b*-PLA BCCP films.⁴¹ This is likely because PDMS and PCL side chains have fast chain dynamics compared to PS and PLA side chains and possible kinetically trapped structures during solvent evaporation can be effectively erased during the subsequent thermal annealing step. The reproducibility of our experiments establishes the basis for reliable model-guided inverse design.”

Supplementary Fig. 16. Reflectance spectra taken by an integrating sphere and bright field optical images of PDMS₆₈¹¹⁰-42%PCL₂₉¹³⁹ photonic films via different solvent vapor annealing. Insets are optical images with size of 0.25 cm × 0.25 cm.

Comment 2.5

The authors may consider clarifying the angular range used in the simulations versus experiments. While the model incorporated angular scans from 0° to 24°, the reflectance spectra were acquired with an integrating sphere. This difference could affect the comparison and should be briefly addressed.

Response 2.5

We appreciate the reviewer’s comment. For the reflectance spectra in both the crystalline and melt states, we use an optical microscope (objective NA = 0.21) coupled with a thermal stage and a fiber-optic spectrometer. The collection angle of the objective is consistent with that used

in the optical model. The lamellar spacing is calculated from the reflectance spectra obtained with this measurement method. We have provided a more detailed description in the Section S2 to clarify the correspondence between the experimental measurements and the simulations. In addition, we have explained all three different setups for spectrum measurements in experimental section and indicated the measurement setup in the captions of all spectra.

[SI, Section S2 on Page 9] “We incorporate angular variation scans from 0° to 24° with 8° step intervals, which is consistent with the measurement angles that is defined by the objective (NA = 0.21) in the customized microspectrometer.”

[Main Text, Lines 450-464 on Page 18-19] “We had three different optical setups to measure the reflectance spectrum of BBCP films. To characterize large-area BBCP films, we employed an integrating sphere (Ideaoptics IS20, sphere size 20 mm) ... We took dynamic spectra and microscopic images during heating-cooling cycles using an optical microscope... To quantify the angle dependence, a custom-built goniometer was employed to measure the angle-resolved reflectance by independently varying the incident and detection angles.”

Response to Reviewer #3

Comment 3.1

The article by D. Yang et al. develops a hybrid experimental–theoretical framework for designing polymeric materials with tunable, precisely controlled colours by adjusting parameters of the polymer architecture. The topic is certainly interesting and of broad relevance to the scientific community and industry for sensing, photonic, and camouflage applications. The underlying theoretical model, based on the work of Zhulina et al. [Macromolecules 53, 534 2582–2593 (2020)], is relevant across the high-segregation limit examined in this work, while the synthesis of the samples and their characterization appear to be carried out with accuracy. The write-up of the article is in-line to the standards of the journal: the manuscript is clear and reports the core findings of the work, whereas the associated details regarding the theoretical model, experimental characterization and sample parameters are appropriately provided in the supporting information. There are, however, a few issues that should be addressed before the manuscript can be recommended for publication. In particular, the optimization procedure based on the four reference samples should be described in greater detail to improve the reproducibility of the study and strengthen the robustness of the proposed framework. Furthermore, some inconsistencies appear in the figures, both in the reported data points and in the corresponding captions, which should be carefully corrected.

Overall, this is an excellent work and once the authors have addressed the points raised herein, the manuscript could be published to the journal, which emphasizes work that is both highly novel and of broad general interest, while also maintaining the highest levels of quality and reproducibility.

Response 3.1

We appreciate the reviewer for the thorough summary of our work and encouraging comments. We will address the concern about the robustness of the colour design model in Responses 3.4-3.5 and some inconsistencies and typos in Responses 3.10-3.11.

Major

Comment 3.2: Stiffening of side chains (α_s) and Eq. (26, SI)

How does α_s affect the domain spacing predicted by eq. 26? I presume it affects \tilde{L}_A , η_A and η_B , since the aforementioned quantities depend on b_s which is affected by crystallinity and thus becomes a function of α_s . Please clarify in the manuscript or the SI.

Response 3.2

We appreciate the reviewer's question. According to Eq. (26), the lamellar spacing d is a function of \tilde{L}_A , η_A and η_B (with A = PDMS and B = PCL or PEO), which can be simplified as

$d_i \sim \tilde{L}_A \left(\text{constant1} + \text{constant2} \times \beta \frac{\eta_B^2}{\eta_A^2} \right)^{-1/3}$ when the chemistry and volume fraction of BBCPs are known. The stiffening parameter is defined as the ratio of the Kuhn lengths of the side chains in the crystalline and melt states, $\alpha_s \equiv b_{s,c}/b_{s,m}$. Side chain A is not crystallizable and $\tilde{L}_A \equiv \left(\frac{4}{\Phi_A^*} \frac{v_{s,A}^2}{(n_{s,A} l_{s,A} b_{s,A})^{1/2}} \frac{\gamma_{A/B}}{k_B T} \right)^{1/3}$ remains the same. The topological parameter, $\beta \frac{\eta_B^2}{\eta_A^2} = \left(\frac{\Phi_B^*}{\Phi_A^*} \right)^{1/2} \left(\frac{n_{s,B} l_{s,B} b_{s,B}}{n_{s,A} l_{s,A} b_{s,A}} \right)^{1/4}$, changes with α_s due to changes in the Kuhn length of side chain B.

Thus, the domain spacing will change with α_s . As shown in Figure R1, the domain spacing decreases gradually with α_s if we use PDMS-*b*-PCL BBCPs for example. When the volume fraction or chain length of PCL increases, the decrease in domain spacing d with α_s becomes more pronounced. We have added revisions to Section S1.3 as:

[SI, Section S1.3 on Page 7] “According to Eq. (26), the lamellar spacing d is a function of \tilde{L}_A , η_A and η_B (with $A = \text{PDMS}$ and $B = \text{PCL}$ or PEO). Considering side chain A does not crystallize and side chain B is crystallizable, we can assume that \tilde{L}_A remains constant and the topological parameter ($\beta \frac{\eta_B^2}{\eta_A^2}$) will change with α_s based on Eq. (19). Therefore, variations in α_s lead to corresponding changes in domain spacing d .”

Figure R1: Temperature-dependent changes in stiffening parameter (α_s) and variation of domain spacing (d/d_0) in samples with different PCL volume fractions (a) and different chain lengths (b). d_0 denotes the domain spacing at $\alpha_s = 1$.

Comment 3.3: Stiffening of side chains (α_s) and phase diagrams

(1) According to Eq. (29), the characteristic regions of the phase diagram, depend on the ratio $\frac{\eta_B}{\eta_A}$. Supposing that the effect of crystallinity is different to η_B and η_A , then the characteristic regions of the phase-diagrams are expected to vary with crystallinity as well.

Is this correct?

(2) Supposing that α_s is the same for the kind-A and -B segments, it might be that the phase diagrams become α_s -independent (according to cancelations arising in Eqs 4-8).

Is this correct? Please check.

If that's the case, do the authors find this reasonable? One would expect that crystallinity would certainly affect the phase diagram, even in cases where it has a similar effect the two species that comprise the copolymer.

(3) Of course, the fact that a single value is adequate for describing the effect very fortunate, but I'm wondering regarding whether the model could be further generalized.

Could the model work with pairs of stiffening parameters ($\alpha_{s,A}$, $\alpha_{s,B}$) applied individually to the kind-A and -B segments, instead to the single value reported in the manuscript ($\alpha_s = 1.5$)? Please elaborate.

Response 3.3

(1) We thank the reviewer for these comments. The functions for phase boundaries (Eq. 29) are independent on Kuhn lengths of side chains and thus crystallinity does not change the diagram based on the SS-SCF theory. However, when the crystallinity of either block varies, the corresponding data point in the phase diagram shifts. Specifically, the x-coordinate (volume fraction of the block) changes due to variations in density, while the y-coordinate (topological parameter, see Eq. 19) changes due to variations in the side-chain Kuhn length. Whether the morphology will change depends on both the direction and magnitude of this shift.

(2) According to Eqs. 4-8, in the symmetric and idealized case where both blocks have the same α_s as well as the same change in density, the effect would be mathematically canceled.

In principle, we can generalize the framework by incorporating two independent stiffening parameters, $\alpha_{s,A}$, $\alpha_{s,B}$, applied individually to the A and B segments. In such a case, the Kuhn length of each block would be rescaled according to its own $\alpha_{s,i}$ ($i = A$ or B). In our present BBCP system, however, block A is amorphous and its Kuhn length is essentially unaffected by crystallization, while block B is crystallizable and exhibits a significant stiffening effect. For this reason, we consider $\alpha_{s,A} \approx 1$, and introduce only a single effective stiffening parameter α_s to capture the change in block B.

(3) We agree that the generalized formulation with ($\alpha_{s,A}$, $\alpha_{s,B}$) is possible and could be useful for copolymers where both blocks undergo crystallization or experience different degrees of stiffening, but in our specific case the single-parameter description is sufficient. Note that a single value α_s is introduced to *phenomenologically* describe the effect of side chain crystallization, with the assumption that the crystallinity is almost the same for all architectures and the interfacial tension between two blocks are not affected by the crystallization. We also

assume that the crystallization only involves the local reorganization of side chains, and the formation of spherulites is suppressed due to the strong confinement of the lamellar structure [ACS Macro Lett. 11, 1085–1090 (2022)]. We have added relevant discussion of the stiffening parameter in the revised supporting information:

[SI, Section S1.3 on Page 7] “We can use a generalized formulation with $(\alpha_{s,A}, \alpha_{s,B})$ for copolymers where both blocks undergo crystallization or experience different degrees of stiffening, but in our specific case the single-parameter description is sufficient. Note that a single value α_s is introduced to phenomenologically describe the effect of side chain crystallization, with several assumptions: (1) the crystallinity is almost the same for all architectures; (2) the interfacial tension between two blocks is not affected by the crystallization; (3) the crystallization only involves the local reorganization of side chains and the formation of spherulites⁵ is suppressed due to the strong confinement of the lamellar structure.”

Comment 3.4: Optimization of α_s and \tilde{L}_A - Technical Details

I cannot find relevant information regarding the optimization of the single α_s parameter. According to lines 173-174 of the main text:

“Incorporating a stiffening parameter of $\alpha_s=1.5$ yields theoretical predictions that align well with experimental domain spacings across all four BBCPs at the crystallized state (Fig. 2e).”, whereas, according to the SI document:

“We adjust α_s for different bottlebrush diblock copolymer systems to achieve the best match between the experimentally observed ratio of domain spacings in the melt and crystalline states, d_c/d_m , and the theoretical prediction given by Eq. (26). It is determined that $\alpha_s = 1.5$ for both PDMS-*b*-PEO and PDMS-*b*-PCL BBCPs.”

Additionally, the following passage in the main text, does not provide sufficient details regarding the optimization procedure of \tilde{L}_A :

“To determine \tilde{L}_A and α_s for the SS-SCF model, we match theoretical calculations to experimental domain spacings of PDMS-211 *b*-PCL films at both melt and crystallized states (black circles in Fig. 3b, Supplementary Fig. 13). This gives $\tilde{L}_A=0.35$ nm and $\alpha_s=1.5$.”

Please add technical information regarding the procedure followed for optimizing α_s and \tilde{L}_A , in order to enhance the reproducibility of the framework and make this work more accessible to the scientific community.

What is the cost function that was optimized? Do the authors employ a specific algorithm for optimizing these free parameters? Could the authors estimate the associated error bar? Are the \tilde{L}_A and α_s parameters optimized simultaneously or one at a time? Is there room for improvement (e.g., see next comment)?

Response 3.4

We thank the reviewer for these detailed comments. In our modified SS-SCF model, the parameter optimization is performed in two steps. First, \tilde{L}_A is optimized in the melt state; then, α_s is optimized in the crystallized state. This stepwise optimization strategy ensures that each parameter retains a clear physical meaning while minimizing the coupling effects between parameters on the fitting results.

For the optimization of \tilde{L}_A , we can simplify Eq. 26

$$d_i = \tilde{L}_A N_A^{2/3} i^{1/3} \left(b_i + g_i(x) \beta \frac{\eta_B^2}{\eta_A^2} \right)^{-1/3} (1+x)^{1/i} \quad (26)$$

into Eq. R1

$$d_i = \tilde{L}_A m \quad (R1)$$

We combined the four experimental points d , obtained from the reflection spectra at the melt state, with the corresponding parameters listed in Table S1 to determine the four values of m . These values were then substituted into Eq. (R1), and the resulting slope was used to extract \tilde{L}_A . This procedure ensures that \tilde{L}_A is determined consistently from multiple experimental data points, accounting for both measured spacings and known polymer parameters. As noted in Figure R2, we have performed a linear fit between the experimentally measured spacing d_{expt} and the theoretically calculated m_{thy} , yielding a slope of 0.35 with a coefficient of determination $R^2 = 0.994$ in the PDMS-*b*-PCL system.

Figure R2: Scatter plots of experimentally measured spacing d_{expt} with theoretically calculated m_{thy} for PDMS-*b*-PCL. The red line is a linearly fitting with a slope of 0.35 and $R^2=0.994$.

After obtaining \tilde{L}_A ($\tilde{L}_A = 0.44$ nm for PDMS-*b*-PEO and $\tilde{L}_A = 0.35$ nm for PDMS-*b*-PCL), we subsequently optimize α_s in the crystallized state. In Eq. (26), the only remaining uncertain parameter is the topological parameter $\beta^{\frac{1}{2}} \frac{\eta_B}{\eta_A}$, which is a function of α_s . By substituting the experimental layer spacings d in the crystallized state, the previously determined \tilde{L}_A , and the parameters from Table S1 into this function, we perform a fitting procedure to determine α_s . The mean squared error between predicted and measured domain spacing is minimized as a function of α_s as shown in Figure R3. It is determined that $\alpha_s = 1.96$ for PDMS-*b*-PEO and $\alpha_s = 1.62$ for PDMS-*b*-PCL BBCPs. These values of α_s are slightly larger than the previous value of 1.5 since we use a more rigorous fitting algorithm. We have added new Supplementary Fig. 2 for PDMS-*b*-PEO and new Supplementary Fig. 17 for PDMS-*b*-PCL, along with a more detailed note in the revised supporting information to clarify the optimization procedure and its rationale.

Figure R3: Minimization of the mean squared error between the predicted and measured domain spacing in the crystallized state determines the stiffening parameter of the side chain (α_s). **a**, PDMS-*b*-PEO; **b**, PDMS-*b*-PCL.

For the optimization of \tilde{L}_A :

[SI, Section S1.3 on Page 6] “Specifically, we compute $m_{thy} = N_A^{2/3} i^{1/3} \left(b_i + g_i(x) \beta \frac{\eta_B^2}{\eta_A^2} \right)^{-1/3} (1+x)^{1/i}$ with the molecular parameters listed in Table S1 and plot them against four measured domain spacing measurements (d_{expt}) that are obtained from the reflection spectra. The characteristic length scale \tilde{L}_A is then extracted from the slope of the linear fit. This procedure ensures that \tilde{L}_A is determined consistently from multiple experimental data points, accounting for both measured spacings and known polymer parameters.”

[Main Text, Lines 177-180 on Page 7] “By fitting the calculated domain spacings to the experimental values at the melt state using Eq. (26) (details in Section S1.3), we determine $\tilde{L}_{PDMS} = 0.44$ nm, in close agreement with the independently calculated $\tilde{L}_{PDMS} = 0.45$ nm.”

For the optimization of α_s :

[SI, Section S1.3 on Page 7] “To obtain the value of α_s , we use the value of \tilde{L}_A obtained from the melt state into Eq. (26), i.e. $\tilde{L}_A = 0.44$ nm for PDMS-*b*-PEO and $\tilde{L}_A = 0.35$ nm for PDMS-*b*-PCL, and then minimize the mean squared error between theoretical prediction and experimental measurements (Supplementary Fig. 7 for PDMS-*b*-PEO and Supplementary Fig. 14 for PDMS-*b*-PCL). It is determined that $\alpha_s = 1.96$ for PDMS-*b*-PEO (Supplementary Fig. 10b) and $\alpha_s = 1.62$ for PDMS-*b*-PCL BBCPs (Supplementary Fig. 17d).”

Supplementary Fig. 3. a, Linear fit between the experimentally measured spacing d_{expt} at the melt state and the theoretically calculated m_{thy} for four PDMS-*b*-PEO samples. The fit yields a slope of 0.44 with $R^2 = 0.904$. **b**, Minimization of the mean squared error between the predicted and measured domain spacing in the crystallized state determines the stiffening parameter of the side chain (α_s) for four PDMS-*b*-PEO.

Supplementary Fig. 4. Optimization of \tilde{L}_A and α_s using four PDMS-*b*-PCL samples. **a**, Reflectance spectra taken by a microspectrometer and bright field optical images of PDMS-*b*-PCL photonic films with different molecular weights and fixed PCL volume fraction (42%) at the melt state. Scale bars are 50 μm in optical images. **b**, Linear fit between the measured spacing d_{expt} at the melt state and the calculated m_{thy} for four PDMS-*b*-PCL samples. The fit yields a slope of 0.35 with $R^2=0.994$. **c**, Theoretical and experimental domain spacings almost collapse on a master line for four PDMS-*b*-PCL BCBPs with different molecular weights at the melt state. **d**, Minimization of the mean squared error between the predicted and measured domain spacing in the crystallized state determines the stiffening parameter of the side chain (α_s) for four PDMS-*b*-PCL samples.

Comment 3.5: Optimization of α_s and \tilde{L}_A - underlying reference data

(1) According to the manuscript the optimization of the aforementioned quantities is based in the 4 reference samples shown in Fig. 3b (or Supplementary Fig. 13b)

“To determine \tilde{L}_A and α_s for the SS-SCF model, we match theoretical calculations to experimental domain spacings of PDMS-*b*-PCL films at both melt and crystallized states (black circles in Fig. 3b, Supplementary Fig. 13). This gives $\tilde{L}_A=0.35$ nm and $\alpha_s=1.5$.” [lines 210 - 213]

According to caption of supplemental Fig 13:

“b, Theoretical and experimental domain spacings almost collapse on a master line for four PDMS-*b*-PCL BBCPs with different molecular weights at the melt state.” the samples are in the melt state. I do not understand how it was possible to optimize as based on these samples. I presume that the authors meant to refer to Fig. 2e, displaying data for the melt and crystalline states. This passage should be clarified to avoid potential confusion for readers.

(2) The data shown in Fig. 3b, illustrate high variability, albeit the trend is consistent. However, supposing that a different set of reference data was chosen for the optimization (e.g., data from $n_{s,PCL} = 38$, which show high variability), one could potentially derive a very different slope in Fig. 3b. What does this imply regarding the robustness of the optimization procedure and the associated predictability of the model?

Response 3.5

(1) We agree that this paragraph needs clarification and we have rewritten this part in the revised manuscript as:

*[Main Text, Lines 234-242 on Page 10] “To determine \tilde{L}_A and α_s for PDMS-*b*-PCL system, we run two steps of optimization in the same manner as for the PDMS-*b*-PEO system. According to Eq. (26), we first run a linear fitting using the experimentally four measured layer spacings d in the melt state (obtained from reflection spectra in Supplementary Fig. 17a), combined with the polymer parameters listed in Table S1. The slope gives $\tilde{L}_A = 0.35$ nm, which is then used to optimize α_s based on layer spacing data in the crystallized state and gives $\alpha_s = 1.62$ (black circles in **Error! Reference source not found.**, Supplementary Fig. 17). This two-step optimization approach ensures that each parameter retains a clear physical meaning while minimizing the coupling between parameters.”*

(2) We agree that selecting a different subset of reference points could affect model parameters, \tilde{L}_A and α_s . After replotting Fig 3b to new Supplementary Fig. 23, we find a great linear trend for samples with different molecular weights but fixed volume fractions and side chain lengths. Variation in volume fraction may lead to the different crystallinity for distinct PCL side chain lengths. In addition, we cannot ignore experimental variations between different batches such as uncertainty in measuring small volumes of Grubb’s catalysts during synthesis. These parameters may all contribute to these variations.

\tilde{L}_A can vary with different subset of reference points, while optimization of α_s is robust. We use 19 data points as reference to optimize α_s with $\tilde{L}_A = 0.35$. We still obtain the same optimal value of 1.62 while the comparison of theoretical and experimental domain spacings shows slight variations (Figure R4). We have added relevant discussions in the revised manuscript.

[Main Text, Lines 252-259 on Page 10-11] “We observe that predicted domain spacings are consistently larger than measured domain spacings for two sample groups (PDMS₆₈-50%PCL₃₈ and PDMS₆₈-42%PCL₂₂), likely due to variation in crystallinity of different PCL

side chain lengths (Supplementary Fig. 23). This deviation may also arise from experimental variations between different batches such as uncertainty in measuring small volumes of Grubb's catalysts during synthesis."

Supplementary Fig. 23. Theoretical and experimental domain spacings almost collapse on a master line for 19 PDMS-*b*-PCL BBCPs (including 15 data points and 4 reference data points).

Figure R4: Optimization of α_s using 19 PDMS-*b*-PCL samples with $\tilde{L}_A = 0.35$. **a**, Minimization of the mean squared error between the predicted and measured domain spacings in the crystallized state as reference 19 data points determines the stiffening parameter of the side chain (α_s). **b**, Comparison of theoretical and experimental domain spacings for 19 PDMS-*b*-PCL BBCPs at crystallized state.

Minor

Comment 3.6: Define BBCP acronym in abstract

Please define the BBCP acronym in the abstract, the first time it is mentioned. E.g.: “Here, we develop a colour design model, enabling inverse design of structural colours in bottlebrush block copolymers (BBCP). The model can quantitatively link BBCP”

Response 3.6

We have made suggested revision in the abstract:

[Main Text, Lines 20-21 on Page 1] “Here, we develop a colour design model, enabling inverse design of structural colours in bottlebrush block copolymers (BBCPs).”

Comment 3.7: Fig. 2, caption – Clarification of the multilayer phase

“.. The coloured region represents the multilayer phase..” What is meant by this? Could it be that the authors refer to the lamellae regime across the phase diagram? Outside this regime (white region) the system finds itself within the cylindrical (C, C') or spherical (S, S') regimes (according to the notation in Zhulina's 2020 paper.) Please clarify.

Response 3.7

As the reviewer noted, the coloured region corresponds to the lamellar regime in the phase diagram, while the white region represents the cylindrical (C, C') or spherical (S, S') regimes. We have clarified this point in the caption of Fig. 2a:

[Main Text, Lines 150-152 on Page 6] “The blue region represents the lamellar phase (L) and the white region represents other structures such as cylindrical (C, C') or spherical (S, S') phases.”

Comment 3.8: Colour swatches

Fig. 3e, depicts a range of colour swatches highlighting the capability of the process to produce a variety of colours as a function of chain architecture. However, there is no information regarding the chain architecture utilized in each case.

The authors should provide this information either in the main text, or in the SI.

I presume that these samples correspond to the samples depicted in Table S2. E.g., they could display an enlarged version of panel 3e and superimpose legends which describe the parameters of each considered chain architecture?

Response 3.8

We have added new Supplementary Fig. 24 in the revised Section S5 on Page 31 to illustrate the colour swatches together with the molecular structure information of corresponding BBCPs.

Supplementary Fig. 24. Colour swatches produced by assembling BBCPs with various chain architectures. Scale bar, 1.5 mm.

Comment 3.9: Supplementary Fig. 15 – Inconsistent data points

The caption of Fig. 15 in SI reads:

“..The data points consist of 22 PDMS-*b*-PCL samples with different chain architectures..” On the other hand, Fig. 15 reports only 9 data points. Please address this inconsistency.

Response 3.9

We thank the reviewer for the comment. Only 9 data points show up in the plot because some samples with identical volume fractions and side chain lengths are overlapping with each other. To avoid this confusion, we have used the shape and color of scattered points to distinguish volume fraction and side chain lengths, as well as adding a new table showing the molecular parameters for all samples (new Supplementary Fig. 19 in SI),

Supplementary Fig. 19. Theoretical phase diagram of PDMS-*b*-PCL BBCPs at the crystallized state based on our modified SS-SCF theory. The data points consist of 22 PDMS-*b*-PCL samples with different chain

architectures and they all fall within the lamellar phase region, matching experimental observations. Some samples with same volume fraction and side chain length occupy the same position in the phase diagram.

Comment 3.10: Supplementary Fig. 19 – Inconsistent data points

The caption of Fig. 19 in SI reads:

“..almost collapse on a master line for 14 PDMS-b-PCL BBCPs with varying backbone lengths.”

On the other hand, Fig. 19 reports 15 data points. The points conform with the plot shown in the main document, which reports 15 + 4 (reference) = 19 points in total. Please address this inconsistency.

Response 3.10

We have corrected the typo in new Supplementary Fig. 23.

[SI, Section S5 on Page 31] “Theoretical and experimental domain spacings almost collapse on a master line for 19 PDMS-b-PCL BBCPs (including 15 data points and 4 reference data points). ...”

Comment 3.11: Grammar, Syntax and Suggestions for change

- Eqs. (1) and (2) in SI report the variable “ N ”, which isn’t defined anywhere. I presume, that’s a typo and N is actually N_b . Please correct this typo and check the files for other instances.
- Lines 34-35: Please change the passage either to: “Quantitative structure-property relationships make inverse design more efficient and predictable.” Or “A quantitative structure-property relationships make inverse design more efficient and predictable”

Response 3.11

We appreciate the reviewer’s comment and have corrected the grammatical errors and carefully proofread the entire manuscript.

The article by D. Yang et al. develops a hybrid experimental–theoretical framework for designing polymeric materials with tunable, precisely controlled colors by adjusting parameters of the polymer architecture. The topic is certainly interesting and of broad relevance to the scientific community and industry for sensing, photonic, and camouflage applications.

The underlying theoretical model, based on the work of Zhulina et al. [Macromolecules 53, 534 2582–2593 (2020)], is relevant across the high-segregation limit examined in this work, while the synthesis of the samples and their characterization appears to be carried out with accuracy.

The write-up of the article is in-line to the standards of the journal: the manuscript is clear and reports the core findings of the work, whereas the associated details regarding the theoretical model, experimental characterization and sample parameters are appropriately provided in the supporting information.

There are, however, a few issues that should be addressed before the manuscript can be recommended for publication. In particular, the optimization procedure based on the four reference samples should be described in greater detail to improve the reproducibility of the study and strengthen the robustness of the proposed framework. Furthermore, some inconsistencies appear in the figures, both in the reported data points and in the corresponding captions, which should be carefully corrected.

Overall, this is an excellent work and once the authors have addressed the points raised herein, the manuscript could be published to the journal, which emphasizes work that is both highly novel and of broad general interest, while also maintaining the highest levels of quality and reproducibility.

Major

1. Stiffening of side chains (a_s) and Eq. (26, SI)

How does a_s affect the domain spacing predicted by eq. 26?

I presume it affects \tilde{L}_A , η_a and η_b , since the aforementioned quantities depend on b_s which is affected by crystallinity and thus becomes a function of a_s . Please clarify in the manuscript or the SI.

2. Stiffening of side chains (a_s) and phase diagrams

●According to Eq. (29), the characteristic regions of the phase diagram, depend on the ratio $\frac{\eta_B}{\eta_A}$. Supposing that the effect of crystallinity is different to η_B and η_A , then the characteristic regions of the phase-diagrams are expected to vary with crystallinity as well.

Is this correct?

●Supposing that a_s is the same for the kind-A and -B segments, it might be that the phase diagrams become a_s -independent (according to cancelations arising in Eqs 4-8).

Is this correct? Please check.

●If that's the case, do the authors find this reasonable? One would expect that crystallinity would certainly affect the phase diagram, even in cases where it has a similar effect the two species that comprise the copolymer.

Of course, the fact that a single value is adequate for describing the effect very fortunate, but I'm wondering regarding whether the model could be further generalized.

Could the model work with pairs of stiffening parameters ($a_{s,A}$, $a_{s,B}$) applied individually to the kind-A and -B segments, instead to the single value reported in the manuscript ($a_s = 1.5$)? Please elaborate.

3. Optimization of a_s and L_A -- Technical Details

I cannot find relevant information regarding the optimization of the single a_s parameter.

According to lines 173-174 of the main text:

“Incorporating a stiffening parameter of $\alpha_s=1.5$ yields theoretical predictions that align well with experimental domain spacings across all four BBCPs at the crystallized state (Fig. 2e).”,

whereas, according to the SI document:

“We adjust α_s for different bottlebrush diblock copolymer systems to achieve the best match between the experimentally observed ratio of domain spacings in the melt and crystalline states, d_c/d_m , and the theoretical prediction given by Eq. (26). It is determined that $\alpha_s = 1.5$ for both PDMS-b-PEO and PDMS-b-PCL BBCPs.”

Additionally, the following passage in the main text, does not provide sufficient details regarding the optimization procedure of L_A :

“To determine \tilde{L}_A and α_s for the SS-SCF model, we match theoretical calculations to experimental domain spacings of PDMS-211 b-PCL films at both melt and crystallized states (black circles in Fig. 3b, Supplementary Fig. 13). This gives $\tilde{L}_A = 0.35$ nm and $\alpha_s = 1.5$.”

Please add **technical information** regarding the procedure followed for optimizing a_s and \tilde{L}_A , in order to enhance the reproducibility of the framework and make this work more accessible to the scientific community.

What is the cost function that was optimized? Do the authors employ a specific algorithm for optimizing these free parameters? Could the authors estimate the associated error bar? Are the L_A and a_s parameters optimized simultaneously or one at a time? Is there room for improvement (e.g., see next comment)?

4. Optimization of a_s and L_A -- underlying reference data

According to the manuscript the optimization of the aforementioned quantities is based in the **4** reference samples shown in Fig. 3b (or Supplementary Fig. 13.b)

“To **determine \tilde{L}_A and α_s** for the SS-SCF model, we match theoretical calculations to experimental domain spacings of PDMS-211 b-PCL films **at both melt and crystallized states** (black circles in Fig. 3b, Supplementary Fig. 13). This gives $\tilde{L}_A = 0.35$ nm and $\alpha_s = 1.5$.” [lines 210 - 213]

- According to caption of supplemental Fig 13:

“**b**, Theoretical and experimental domain spacings almost collapse on a master line for four PDMS-b-PCL BBCPs with different molecular weights **at the melt state**.”

the samples are in the melt state. I do not understand how it was possible to optimize a_s based on these samples.

I presume that the authors meant to refer to **Fig. 2e**, displaying data for the **melt and crystalline states**. This passage should be clarified to avoid potential confusion for readers.

- The data shown in Fig. 3b, illustrate high variability, albeit the trend is consistent. However, supposing that a difference set of reference data was chosen for the optimization (e.g., data from $n_{s,PCL} = 38$, which show high variability), one could potentially derive a very different slope in Fig. 3b. What does this imply regarding the robustness of the optimization procedure and the associated predictability of the model?

Minor

5. Define BBCP acronym in abstract

Please define the BBCP acronym in the abstract, the first time it is mentioned. E.g.:

“Here, we develop a colour design model, enabling inverse design of structural colours in bottlebrush block copolymers (BBCP). The model can quantitatively link BBCP”

6. Fig. 2, caption – Clarification of the multilayer phase

“.. The coloured region represents the multilayer phase..”

What is meant by this? Could it be that the authors refer to the lamellae regime across the phase diagram? Outside this regime (white region) the system finds itself within the cylindrical (C, C') or spherical (S, S') regimes (according to the notation in Zhulina's 2020 paper.) Please clarify.

7. Fig. 3e – Colour swatches

Fig. 3e, depicts a range of colour swatches highlighting the capability of the process to produce a variety of colors as a function of chain architecture. However, there is no information regarding the chain architecture utilized in each case.

The authors should provide this information either in the main text, or in the SI.

I presume that these samples correspond to the samples depicted in Table S2. E.g., they could display an enlarged version of panel 3e and superimpose legends which describe the parameters of each considered chain architecture?

8. Supplementary Fig. 15 – Inconsistent data points

The caption of Fig. 15 in SI reads:

“..The data points consist of **22** PDMS-b-PCL samples with different chain architectures..”

On the other hand, Fig. 15 reports only **9** data points. Please address this inconsistency.

9. Supplementary Fig. 19 – Inconsistent data points

The caption of Fig. 19 in SI reads:

“..almost collapse on a master line for **14** PDMS-b-PCL BBCPs with varying backbone lengths..”

On the other hand, Fig. 19 reports **15** data points. The points conform with the plot shown in the main document, which reports 15 + 4 (reference) = 19 points in total.

Please address this inconsistency.

10. Grammar, Syntax and Suggestions for change

- Eqs. (1) and (2) in SI report the variable “N”, which isn’t defined anywhere. I presume, that’s a typo and N is actually N_b . Please correct this typo and check the files for other instances.

- Lines 34-35:

Please change the passage either to:

“~~A~~ ~~q~~Quantitative structure-property relationships makes inverse design more efficient and predictable.”

Or

“A quantitative structure-property relationships ~~s~~ makes inverse design more efficient and predictable”